# Test case sampling optimization for safety validation of automated driving systems

Chen Qian ®[1,5], Jingbin Xu ®[2,5], Xin Xing ®[3] & Feng Guo ®[3,4] ✉

Testing and validating automated driving systems require carefully designed test cases that capture the complexity of real-world driving conditions. However, the inherent complexity of driving environments and the rarity of safety-critical situations pose significant challenges to developing reliable and efficient validation frameworks. This paper addresses these issues by selecting appropriate test cases from the largest-scale naturalistic driving study. We introduce a Kernel Test Case Sampling method, which selects cases satisfying two key criteria: representativeness, ensuring alignment with real-world scenarios, and coverage, capturing high-risk corner cases. To demonstrate the proposed method, it is applied to large-scale naturalistic driving study data. By selecting a limited number of cases, the method effectively captures long-tailed scenarios while approximating the distribution of naturalistic driving conditions. The sampling framework also enables robust accident-rate estimation, thereby ensuring fair comparisons across human driving performance and multiple systems. The proposed method supports standardized and scalable automated driving system safety validation, facilitating accelerated development and deployment while building public trust and regulatory confidence.

The widespread adoption of automated driving systems (ADS) hinges on a reliable framework for safety validation[1]. Researchers, industry professionals, and regulators invested substantial effort[2,3] in developing test cases that are then evaluated through various validation venues, such as test tracks[4] or simulators[5]. However, current approaches rely on heuristic rules to select well-defined test cases[6], such as car following scenarios, which do not fully capture the unexpected edge cases that can arise in real-world driving[7,8]. Furthermore, as ADS safety standards evolve, reliance on static testing procedures can become increasingly inadequate and lead to potential biases[2]. A more strategic approach to test case selection − carefully choosing from the millions of possible driving scenarios − can help mitigate such biases in the testing process. Once testing is completed, it is essential to interpret the results using well-defined safety metrics, such as crash rates and the number of disengagements. Since most transportation regulations depend on human-driving benchmarks to develop safety

countermeasures[9], the deployment of comparable human-driving safety metrics is crucial to build public trust, ensure regulatory compliance, and facilitate the wider deployment of ADS[10].

The selection of test cases for ADS safety validation involves identifying and prioritizing critical scenarios that challenge the system's capabilities within its operational design domain. Existing research often relies on surrogate models to approximate the driving behaviors of background vehicles or assumes overly simplified driving actions, such as constant deceleration maneuvers[11]. Notably, these approaches may not fully represent real-world conditions and could have limited coverage of the parameter space. As shown in Fig. 1a, these assumptions can diverge significantly from actual on-road conditions and limit the validity of test scenarios. Validating safety is complicated by the challenges of the curse of dimensionality and the curse of rarity, as highlighted in a recent publication[12]. These challenges suggest that a massive number of test cases would be required to comprehensively

[1]Dalian University of Technology, School of Economics and Management, Dalian, China. [2]Dalian University of Technology, School of Mechanical Engineering, Dalian, China. [3]Virginia Tech, Department of Statistics, Blacksburg, VA, USA. [4]Virginia Tech Transportation Institute, Blacksburg, VA, USA. [5]These authors contributed equally: Chen Qian, Jingbin Xu. ✉e-mail: feng.guo@vt.edu

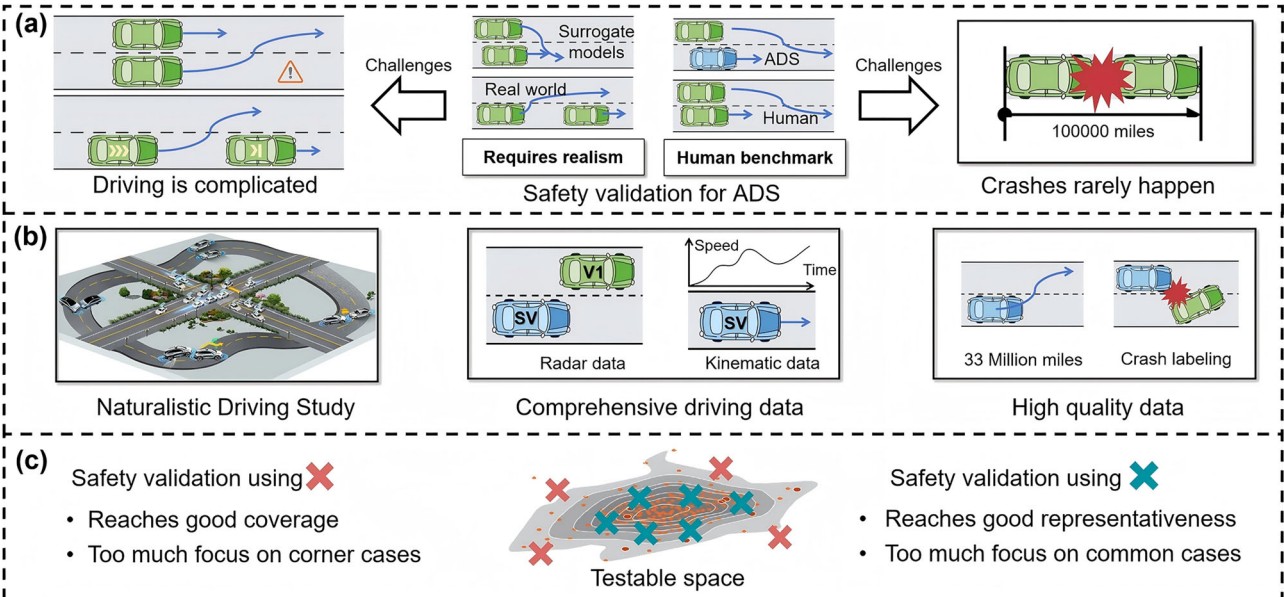

**Fig. 1 | Schematic diagram of test case selection for safety validation of automated driving systems. a** Overview of the test case selections for ADS safety validation. **b** The study utilizes naturalistic driving study data from SHRP2, which provides high-quality, comprehensive driving information. SHRP2 NDS collected data from six U.S sites, providing an unbiased benchmark for human driver driving data. **c** Selected test cases guarantee both representativeness and coverage for effective validation.

cover the full spectrum of the testable driving space. However, in practice, the number of selected test cases must be limited, as replicating or reconstructing each test case through downstream validation venues is costly. The test track approach requires test cases to be replicated using strikeable surrogate vehicles to simulate various elements, including location and driving dynamics such as velocity[4]. In the simulation setup, reconstructing each test case is time-consuming, as it requires precisely recreating every potential risk factor by independent third parties to mitigate potential conflicts of interest[13]. Optimizing the selection of test cases to support an efficient test regimen is therefore essential.

To construct the test case pool for safety validation, we adopt the largest-scale naturalistic driving study (NDS), the Second Strategic Highway Research Program (SHRP2)[14]. The SHRP2 NDS follows a well-designed experimental plan[14,15] to capture nationwide, population-level U.S. driving characteristics, as shown in Fig. 1b. The SHRP2 NDS is a national database representing key traffic safety characteristics[16] and various operating domains[17]. The advanced data acquisition systems record comprehensive driving data, roadway information, and driver behaviors, which are essential for capturing the parametric driving parameters under real-world traffic conditions. Fig.2a illustrates the construction of the test case pool, in which 0.3 million cases are sampled from the SHRP2 NDS dataset. Each sample represents a specific driving scenario that describes the temporal development of consecutive scenes that include relevant static and dynamic elements. Each sample spans 15 s and includes 48 features extracted from radar and kinematic data, capturing driving dynamics, environmental factors, and interaction behaviors.

Two key criteria, representativeness and coverage, are introduced to evaluate the proposed test case sampling method, compared to the state-of-art method, to support comprehensiveness of the test case pool. Representativeness ensures that the selected cases, taken together, reflect the realism of actual driving situations observed on public roads (as derived from the SHRP2 NDS). Using these cases, we can create a test set that realistically mirrors how often different driving scenarios occur, enabling more meaningful and credible safety evaluations. As shown in Fig. 1c, a small number of strategically selected green points around the core of the testable space can effectively

represent the majority of routine driving scenarios, so as to reduce the total number of required cases. Coverage ensures that selected cases include diverse driving scenarios, particularly those that are less frequent yet carry safety-critical information and might otherwise be missed, marked in red in Fig. 1c. For example, a sudden cut-in by a lead vehicle at high relative speed on a curved highway exit ramp is rare in naturalistic driving but highly critical for ADS safety. These regions are prone to unexpected failures, and their inclusion helps prevent overlooked vulnerabilities in ADS validation. Both criteria are essential for a reliable and effective validation process, ensuring that the ADS is tested comprehensively in both common and rare situations.

In the statistics and machine learning domain, various approaches have been proposed to optimize case selection. Uniform sampling[18] ensures an even spatial distribution but neglects low-density, high-risk regions critical for safety validation. The support point algorithm[19] uses convex-concave methods to preserve the statistical properties of the dataset, ensuring representativeness, but it struggles to capture rare, high-risk scenarios. Similarly, SPARTAN[20] employs optimal transport theory to enhance coverage through space-filling properties, but it does not always match real-world distributions. Approaches that focus on underrepresented data points[21] or low-probability selections[22] can effectively target high-risk areas but may lack representativeness. However, none of the existing methods optimizes both representativeness and coverage simultaneously.

We propose the Kernel Test Case Sampling (KTCS) method to simultaneously achieve the properties of representativeness and coverage through a dual optimization approach using kernel methods, as illustrated in Fig. 2. Coverage optimization aims to minimize the information potential (IP)[23] by framing test case selection as an optimization problem. Since directly solving this problem is NP-hard, we employ a stochastic sampling approach that assigns importance measures to each data point using a neural network, followed by Pareto-order sampling to ensure comprehensive coverage of the testable space. The goal of representativeness optimization is to minimize the maximum mean discrepancy (MMD)[24], ensuring that the selected cases closely match real-world driving distributions. To achieve this, an attention mechanism is introduced that assigns distribution-alignment weights to the selected cases[25,26], adjusting the selected subset so that

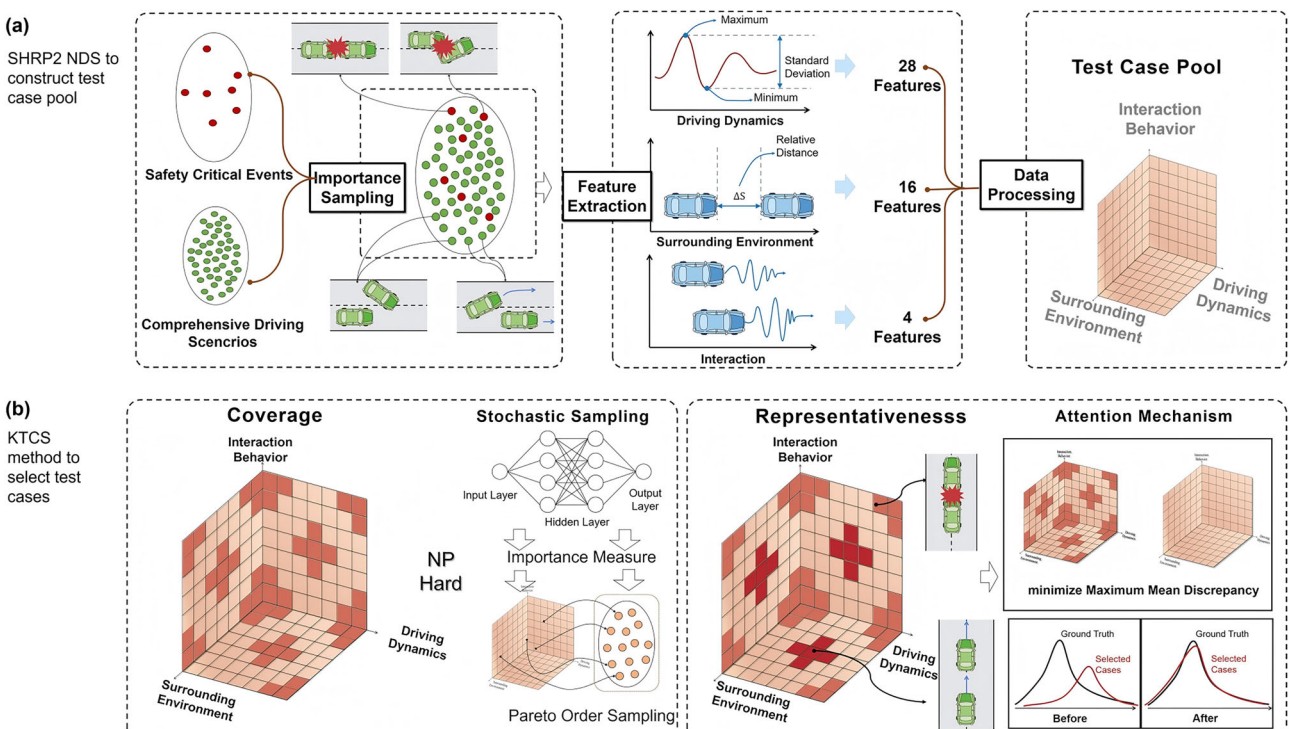

**Fig. 2 | Overview of the test case selection framework for ADS safety validation. a** The test case pool is constructed from the SHRP2 NDS dataset using importance sampling, with extracted features on driving dynamics, environmental factors, and interaction behaviors organized into a structured dataset. **b** Test cases are selected through a two-step process: Stochastic sampling for coverage ensures comprehensive representation of the testable space; Attention mechanism for representativeness assigns weights to align selected cases with real-world scenario distributions.

its empirical distribution matches that of the full candidate test case pool. A larger weight assigned by the attention mechanism indicates that the corresponding selected case contributes more to approximating the underlying distribution of the candidate test-case pool. This process ensures that the selected cases accurately represent the distribution of real-world driving scenarios. We establish the optimal properties of the proposed method through theoretical proof and validate its performance through extensive Monte Carlo simulations.

To demonstrate the interpretation of the test results using selected cases, a measure called Scaling Risk (SR) is proposed to quantify the safety level of an ADS system's performance based on the success or failure of each case, where a case is labeled a success if the SUT completes it without a traffic crash, and a failure if the ADS does not complete it safely and a crash occurs. SR reflects how much more risk the ADS carries compared to human drivers in terms of being involved in traffic crashes. For instance, a score of 2.5 indicates that the ADS system under test is 2.5 times riskier than human drivers.

Addressing the challenge of selecting optimal test cases for ADS safety validation, the proposed framework leverages a small yet informative subset of scenarios drawn from large-scale nationwide NDS data. By strategically selecting test cases that are theoretically guaranteed to be representative of real-world driving conditions, the framework provides a principled foundation for ADS safety validation. Leveraging large-scale NDS data and the proposed KTCS selection method, our approach selects test cases that simultaneously satisfy representativeness and coverage while preserving the realism of real-world driving behavior. The framework further enables reliable performance comparisons between an ADS and human drivers by anchoring evaluation to population-level crash rates and using attention-based, interpretable weights to quantify each test case's contribution to distribution alignment.

## Results

### Data

A systematic sampling scheme is used to construct the test case pool based on traffic crash events and normal driving segments extracted from SHRP2 NDS, ensuring a close approximation of population-level driving characteristics. The test cases are categorized into two groups: normal driving and safety-critical events. Each case consists of a 15-s driving segment, treated as a driving scenario. The selection of normal driving cases follows a two-stage stratified design that preserves the original representation of the driver population and mitigates selection bias across drivers[15]. First, the number of normal driving segments is determined proportionally by the total driving time of each driver. Then, a preset number of driving segments is randomly sampled from each driver's data. In total, about 300,000 randomly selected 15-s driving segments were chosen, covering approximately 50 million meters of driving from the SHRP2 NDS database. Safety-critical events are selected from the full population of crash data in the SHRP2 NDS. Using a validated crash identification method[27], all crash events are identified and incorporated into the candidate test-case pool. Each crash event case consists of a 10-second driving segment leading up to the collision and a 5-s segment following it. Safety-critical situations include severe level-1 traffic accidents and level-2 police-reportable crashes from the SHRP2 NDS database[14]. To ensure that test cases are relevant to ADS capabilities, single-vehicle conflicts were excluded because, when no other moving vehicles are present, the radar cannot provide the surrounding environment data needed to compute surrogate risk metrics[27]. Besides, single-vehicle conflicts in human drivers primarily result from human errors, such as distracted driving, while ADS are shown to mitigate or avoid.

Data preprocessing is conducted by retaining only cases that include both radar and kinematic data. Additionally, the following criteria are applied to ensure the test cases reliably simulate real-world scenarios. (1) Average speed: The average speed of the 15-s driving

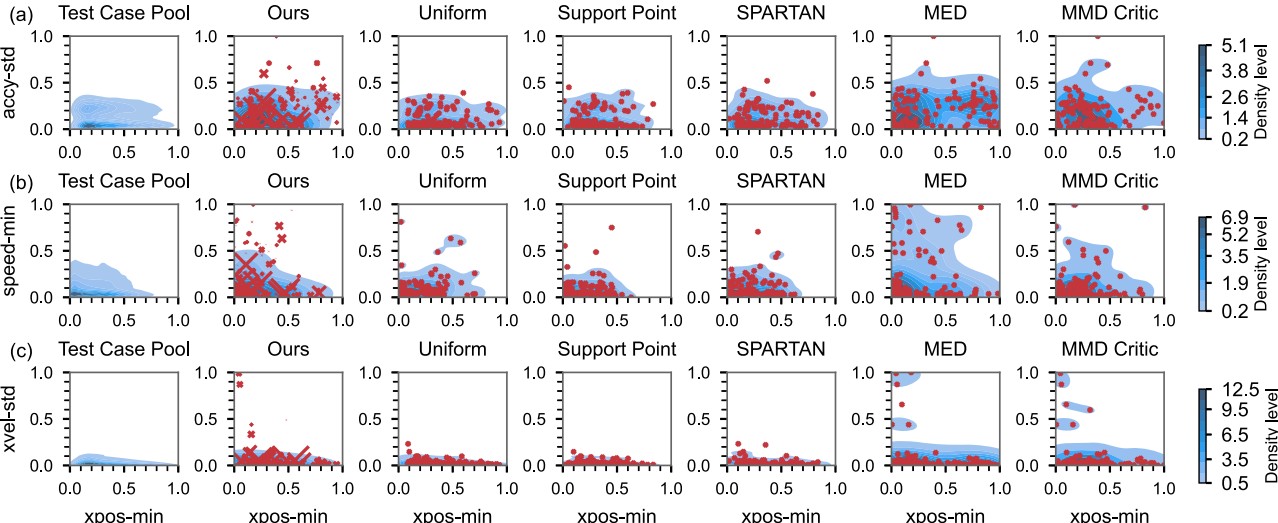

**Fig. 3 | Comparison of density distribution based on points selected by different methods. a** Joint distribution of features *accy-std* and *xpos-min*. **b** Joint distribution of feature *speed-min* and *xpos-std*. **c** Joint distribution of feature *xvel-std* and *xpos-min*. The blue area represents the estimated density distribution, with darker shades indicating higher density. Red points mark the selected cases, and the size of each point reflects the weight of that case.

segment must exceed 5 mph, as previously adopted[14]. (2) Maximum speed: The maximum speed must be less than 35 m/s (80 mph), which corresponds to the speed limit on most U.S. roads. (3) Longitudinal distance: The maximum longitudinal distance to the nearest vehicle must be under 60 meters, as radar detection accuracy decreases beyond this range[28]. In total, 55,920 normal driving segments and 90 crash events are used for the analysis.

## Experiment Setup

Each case within the test case pool comprises two levels of information: kinematics data and radar data. The kinematics data describe the driving state of the subject vehicle (SV), such as the driving speed, while the radar data capture the status of surrounding vehicles. The following set of features is extracted from the driving test case. (1) Driving dynamics: Captures the characteristics of the SV's driving behavior. (2) Surrounding environment: Represents the spatial distribution and relative driving dynamics of nearby vehicles at a specific moment. (3) Driving interactions: Describes the temporal evolution of the driving scenario, accounting for interactions with surrounding vehicles.

Driving dynamics features are extracted using four types of kinematic data: SV's longitudinal acceleration (*accx*), lateral acceleration (*accy*), horizontal acceleration (*accz*), and driving speed (*speed*). For each type of kinematic data, seven statistical features—initiation (*init*), maximum (*max*), minimum (*min*), mean (*mean*), standard deviation (*std*), skewness (*skew*), and kurtosis (*kurt*) - are calculated to capture initial status, extreme conditions, average driving levels, volatility, frequency and intensity of abrupt maneuvers. For example, *speed-min* represents the minimum driving speed in the 15-second driving segment. This feature extraction process results in $7 \times 4 = 28$ kinematic-based features.

The SHRP2 radar system provides relative distance and velocity data for surrounding vehicles in both longitudinal and lateral directions. Features are extracted using the closest vehicle to characterize the surrounding environment[1], including average (*mean*), maximum (*max*), minimum (*min*), and initial (*init*) values for relative longitudinal position (*xpos*), lateral position (*ypos*), longitudinal velocity (*xvel*), and lateral velocity (*yvel*). This results in 16 features reflecting the static environment. An additional 4 features are calculated to assess driving interactions, consisting of the standard

deviation for each radar-related metric (*xpos-std*, *ypos-std*, *xvel-std*, and *yvel-std*).

The feature extraction process generates 48 features: 28 kinematic-based features for driving dynamics, 16 radar-based features for the static environment, and 4 features for driving interactions. Detailed formulas and feature names are provided in the Supplementary Table 4. All features are normalized to [0, 1] for uniform scaling and fair comparisons across 48 dimensions.

## Performance comparison

Define *M* as the number of selected cases and *N* as the total instances in the test case pool. We set $M = 0.5\sqrt{N}$, resulting in 118 selected cases for our analysis. We compare our method against five state-of-the-art approaches: (1) Minimum Energy Design (MED):[22] Sequentially selects data points based on density estimation using a greedy algorithm. (2) MMD-Critic:[21] Identifies prototypes and criticisms within the data using kernel methods. (3) The Support Point (SP):[19] Minimizes energy distance to select representative points. (4) SPARTAN:[20] Leverages optimal transport theory to ensure uniform data coverage. (5) Uniform Sampling (Nystrom Sampling):[29] Randomly selects cases uniformly. Each method is applied to the test case pool to select 118 cases.

Figure 3 compares the sampling representativeness performance of the proposed method with state-of-the-art methods using visualization of the statistical density distribution. The density distributions are generated using kernel density estimation, with darker blue indicating higher density regions. Figure 3a shows the joint distribution of *xpos-min* and *accy-std*. The feature *accy-std* measures the volatility of lateral control, while *xpos-min* represents the minimum relative longitudinal distance to surrounding vehicles. These two features provide insights into lateral stability and proximity to nearby vehicles. Our method effectively captures the negative relationship between these features and covers the tail regions of the distribution. The attention mechanism ensures that the final distribution closely aligns with the original test case pool. Other methods do not reflect this negative relationship, where unstable lateral movements are more likely to occur in crowded or congested environments. MED overly emphasizes corner cases, resulting in a biased estimate of overall distributions. MMD-Critic struggles to balance central modes and corner cases, leading to discrepancies in the central region of the test case pool. Fig. 3b shows the joint density distribution of features *speed-min* and

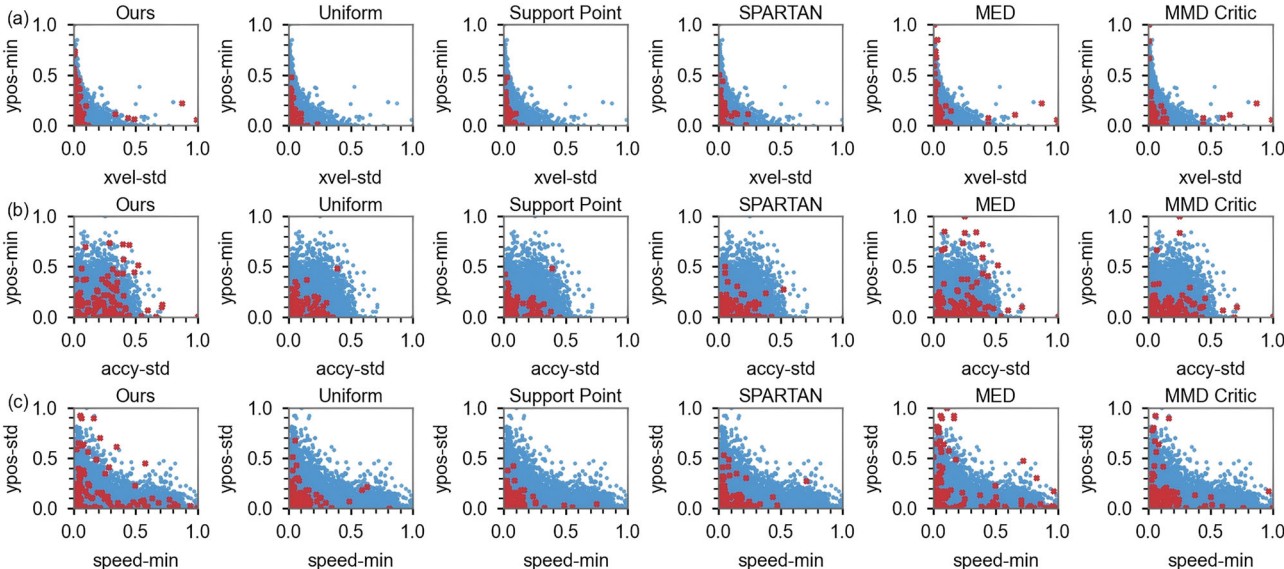

**Fig. 4 | Comparison of selected points across different methods. a** Scatter plot of selected points on the features *ypos-min* and *xvel-std*. **b** Scatter plot of selected points on the features *ypos-min* and *accy-std*. **c** Scatter plot of selected points on the features *ypos-std* and *speed-min*. The blue dots represent the original test case pool, while the red dots indicate the selected points using different methods.

**Table 1 | Sampling performance comparison using Empirical Hellinger distance (H-Dist), Empirical Absolute Distribution Difference (ADD), Average L1 and L2 Pairwise Distances for $M = 0.5\sqrt{N}$**

| Method | IP↓ | MMD↓ | H-Dist ↓ | ADD ↓ | Avg-L1 ↑ | Avg-L2 ↑ |
|---|---|---|---|---|---|---|
| Ours | 0.067 | 0.020 | 0.018 | 1.007 | 8.175 | 1.879 |
| MED | 0.034 | 0.318 | 0.365 | 3.747 | 8.965 | 2.072 |
| MMD-Critic | 0.223 | 0.038 | 0.108 | 1.669 | 5.963 | 1.402 |
| SP | 0.360 | 0.003 | 0.027 | 0.997 | 3.549 | 0.844 |
| Uniform | 0.391 | 0.006 | 0.003 | 0.437 | 3.923 | 0.926 |
| SPARTAN | 0.414 | 0.004 | -0.005 | 0.364 | 3.796 | 0.887 |

Note: ↑ indicates that higher values are better, while ↓ indicates that lower values are better. H-dist is an empirical distance measure that can take negative values[20].

*xpos-std*, which reflect the ability of the SV to adapt its behavior to external conditions. SP and SPARTAN fail to cover high-risk corner cases in low-density regions, while MED and MMD-Critic over-emphasize these cases. A similar pattern is observed in Fig. 3c. Uniform sampling evenly distributes points across the domain, which lacks realism. Our method achieves the best approximation of corner cases, as the attention mechanism assigns appropriate weights to high-risk scenarios while preserving the overall distribution shape.

Figure 4 evaluates the sampling coverage performance of the proposed method compared to state-of-the-art approaches using scatter diagrams. The blue dots represent the original test case pool, and the red points indicate the selected cases for each method. The Uniform, SP, and SPARTAN approaches primarily select cases centered around the mode of the joint distributions, resulting in limited coverage across the entire data space. MMD-Critic selects certain long-tail cases located far from the center of the investigated joint distributions, but it fails to provide comprehensive coverage of the entire long-tail region, as evidenced by the large blue area not spanned by the red points in Fig. 4b, c. While MED demonstrates some capacity to cover long-tailed cases, our method selects points more evenly distributed, particularly near the outer limits of the feasible parameter space. As

shown in Fig. 4c, MED-selected points cluster tightly in the tail, with many exhibiting nearly identical values of *ypos-std* (lateral-distance variability) and *speed-min*, thereby offering limited new information. In contrast, our method distributes points throughout both the tail and interior regions, reducing redundancy and enhancing test efficiency.

After visualizing the sampling capacity, we introduce six measures to theoretically verify the statistical realism of the sampling distribution. The results are summarized in Table 1. The first two metrics, IP and MMD, are the optimization objectives of our algorithms. Smaller IP values indicate better coverage, while smaller MMD values indicate better representativeness[24]. To provide a more comprehensive assessment, we employ four additional metrics. To begin with, representativeness is evaluated by assessing how well the density estimation constructed from the selected cases approximates the test case pool. Given the high-dimensional nature of features, we adopt two metrics to assess overall similarity: one for joint distributions and one for marginal distributions. These metrics are the empirical Hellinger distance (H-Dist)[20] and the Absolute distribution difference (ADD)[30]. Smaller H-Dist and ADD values indicate better approximation capabilities for joint distributions and marginal distributions, respectively. Define the constructed density estimation of the subselected points as $\hat{p}(\cdot)$, and the marginal density estimation for the $j$th dimension as $\hat{p}_j(\cdot)$. Similarly, let the density estimations from the candidate case pool be $p(\cdot)$, with the marginal density for the $j$th dimension represented as $p_j(\cdot)$. The H-Dist and ADD are defined as follows:

$$\text{H}-\text{Dist} = 1 - \frac{1}{N}\sum_{i=1}^{N}\sqrt{\hat{p}(\boldsymbol{x}_i)/p(\boldsymbol{x}_i)} \quad \text{ADD} = \frac{1}{N}\sum_{j=1}^{d}\sum_{i=1}^{N}|\hat{p}_j(\boldsymbol{x}_i) - p_j(\boldsymbol{x}_i)|$$

$$(1)$$

The results of ADD and H-Dist are reported in Table 1.

Next, coverage is assessed through diversity evaluation using pairwise differences between selected points. We follow the widely used method of calculating the average pairwise distances[31]. Larger distance values reflect greater diversity, indicating that the selected samples are more likely to include edge cases and critical regions for safety validation. The L1 and L2 distances are used as the discrepancy

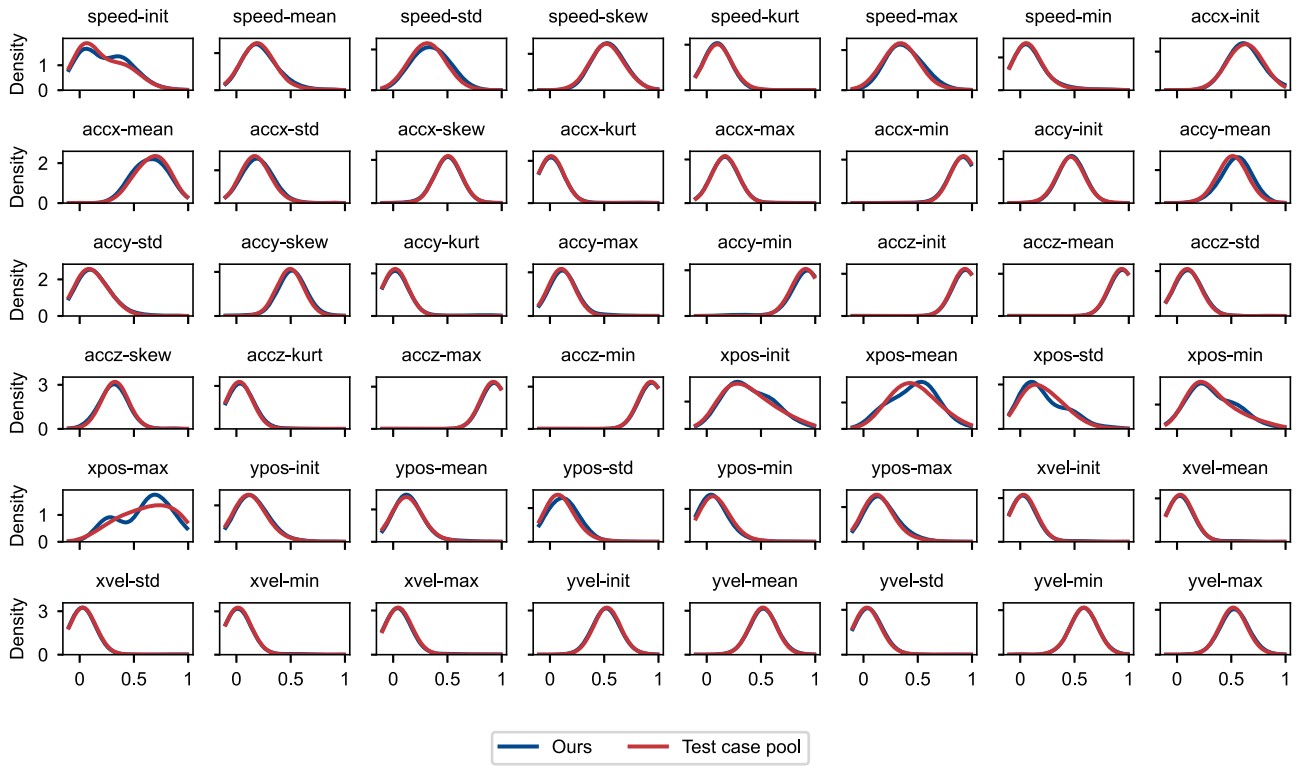

**Fig. 5 | Comparison of density distributions across all 48 features.** The red line represents the density distribution of the entire test case pool, while the blue line represents the density distribution of the 118 cases selected by our method.

measure, defined as follows:

$$\mathrm{Avg-L1} = \frac{1}{M(M-1)} \sum_{i=2}^{M} \sum_{j \neq i, j=1}^{M} \| \mathbf{z}_i - \mathbf{z}_j \|_1$$

$$\mathrm{Avg-L2} = \frac{1}{M(M-1)} \sum_{i=2}^{M} \sum_{j \neq i, j=1}^{M} \| \mathbf{z}_i - \mathbf{z}_j \|_2 \tag{2}$$

where $\| \mathbf{z}_i - \mathbf{z}_j \|_1$ and $\| \mathbf{z}_i - \mathbf{z}_j \|_2$ represent the L1 and L2 distances, respectively, between points $\mathbf{z}_i$ and $\mathbf{z}_j$. The results are summarized in Table 1, where higher values for Avg-L1 and Avg-L2 indicate a greater likelihood of covering corner cases[22].

In summary, Table 1 shows our method achieves competitive scores on both IP and MMD, demonstrating balanced performance across these metrics. Our method outperforms MMD-Critic by achieving lower values on both measures. MED fails to meet the representativeness objective, as its MMD is substantially higher than that of all other methods. Although SP-Uniform and SPARTAN achieve slightly lower MMD values, they produce substantially higher IP scores (all above 0.360). While the points sampled by MED exhibit diverse characteristics, its density estimation fails to align with the test case pool. Uniform sampling, SP and SPARTAN struggle to maintain adequate coverage. Among the five comparison methods, only MMD-Critic attempts to balance representativeness and coverage; however, its performance falls short across all metrics compared to our method.

**Statistical realism**

In this section, we evaluate whether our method's performance meets the requirements for effective test case selection. Specifically, we compare the density distribution generated by the 118 cases selected by our method with the distribution derived from the entire test case pool. Fig. 5 illustrates this comparison for each of the 48 features. The red line represents the density distribution of the entire test case pool,

while the blue line represents the distribution generated by the cases selected by our method. Our method closely approximates the ground truth density distribution across individual dimensions, demonstrating its ability to reflect real-world driving scenarios realism. Furthermore, the consistency observed across all 48 features underscores the robustness of our approach in preserving the statistical properties of the test case pool.

Similarly, we quantitatively compare the per-feature discrepancy across all methods using a heatmap in Fig. 6, where the per-feature discrepancy is measured by $ADD_j = \frac{1}{N} \sum_{i=1}^{N} |\hat{p}_j(\mathbf{x}_i) - p_j(\mathbf{x}_i)|$. Here, $ADD_j$ represents the contribution of feature $j$ to the overall ADD. The global ADD reported in the main paper is the sum of these 48 per-feature discrepancy values. Figure 6 consists of eight sub-panels: the first row displays all kinematics-based features, and the second row shows all radar-based features. Darker cells indicate larger discrepancies, corresponding to lower statistical realism.

To evaluate the coverage achieved by our method, we assess its ability to capture corner cases, as illustrated in Fig. 7. For each test case, we compute its multidimensional distance from the centroid of the candidate test case pool. Larger distances indicate a greater likelihood of being a corner case. Corner cases, characterized by long-tailed characteristics, lie far from the distribution's mode and provide critical information for safety validation. Figure 7a, b shows the relative distances of each case from the center of the test case pool. These distances provide a quantitative measure of how far a case deviates from the core distribution.

Using thresholds of 95% and 99.99%, we identify cases with distances greater than the specified proportion of all cases in the pool. For instance, the 99.99% threshold highlights cases that lie further into the extreme tail of the distribution. Of the 118 cases selected, 77 fall beyond the 99% threshold, and all 6 cases beyond the 99.99% threshold are captured, demonstrating the method's strong ability to cover the most extreme long-tailed driving scenarios. The method

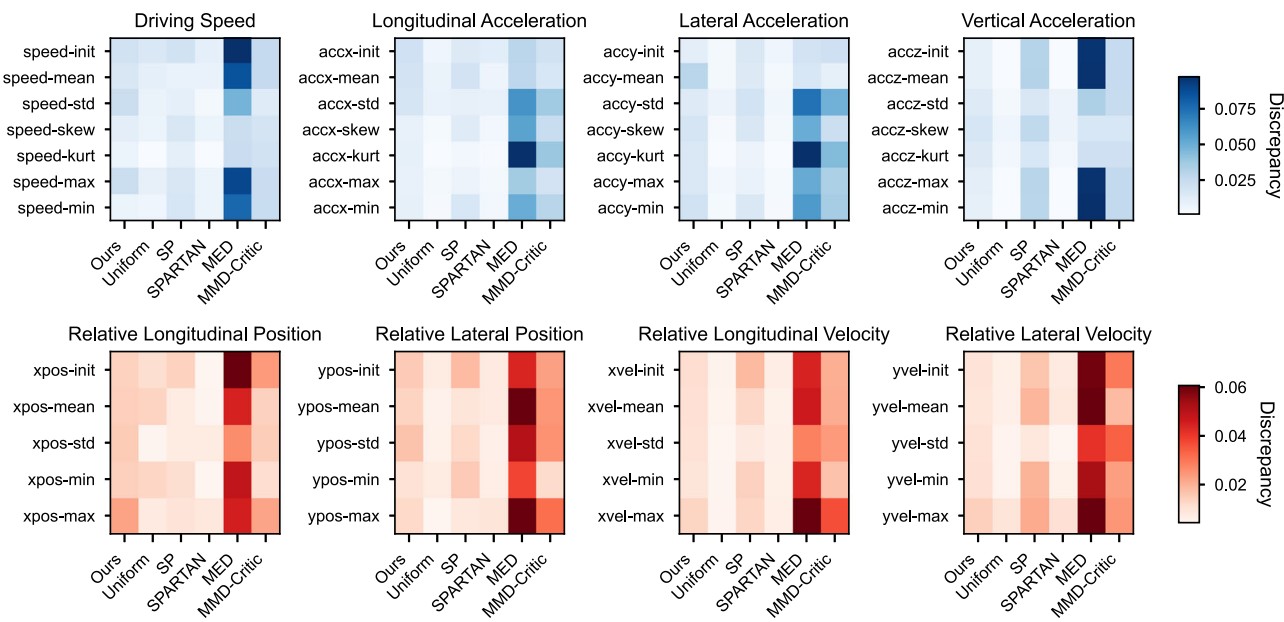

**Fig. 6 | Heat-map of per-feature discrepancies across different methods.** The first row presents results for kinematics-based features, while the second row presents results for radar-based features.

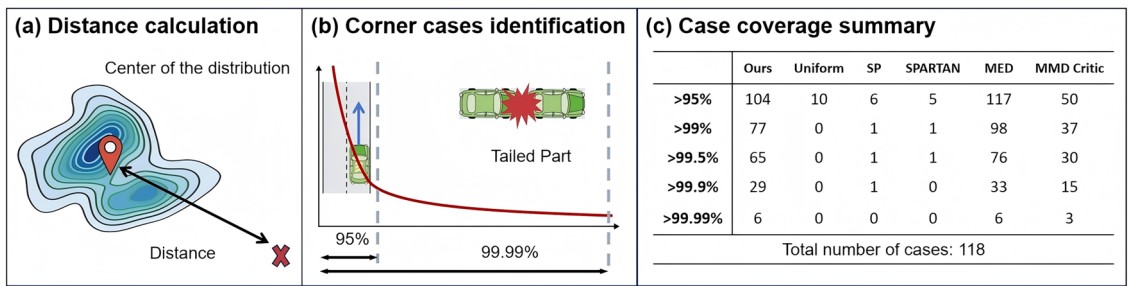

**Fig. 7 | Schematic diagram of coverage evaluation. a** Illustration of distance calculation for the central density region. **b** Identification of corner cases based on different quantile thresholds. **c** Summary of corner case coverage across varying quantile thresholds.

effectively represents the central density region while systematically capturing rare long-tailed events, as shown in Figs. 5, 7.

Figure 7c presents a quantitative comparison of long-tail scenario coverage across all baseline methods. The results show that Uniform, SP, and SPARTAN offer limited coverage of long-tail scenarios, as only a few of their selected cases exceed the 99% percentile. The MED method captures the tail almost exclusively, with 117 of 118 selected cases exceeding the 95% percentile. Such strong emphasis on extreme values limits its ability to represent the full distribution, thereby reducing its overall statistical representativeness.

**Demonstration**

In this section, we introduce a safety measure, namely Scaling Risk, to quantify the safety level of an ADS by comparing its performance in selected testable cases against a human driving benchmark. By using a limited number of test cases, we avoid the need for billions of miles of driving data[32], enhancing validation efficiency without sacrificing reliability. This approach is supported by the theoretically validated sampling mechanism discussed in the previous section, which ensures the selection of the most representative and safety-relevant test cases.

Given a driving system under test (SUT), it is evaluated on a set of 118 selected test cases. For each test case $j$, we define a binary indicator $I_j \in {0, 1}$, where $I_j = 1$ indicates that the SUT passed the case, and $I_j = 0$ indicates a failure (e.g., a crash). To evaluate the safety performance of the SUT, we introduce Accident-Rate-SUT, a metric that converts the

binary test outcomes (pass/fail) of the representative test cases into a real-world, safety-oriented measure. This metric is motivated by the *golden rule* in automated driving system (ADS) safety research: accident rate, defined as the number of crashes divided by the total driving distance[1,33]. The exposure distance, $d_j$, for each case is calculated as the product of the average speed and the fixed test duration (15 s). Our method employs an attention mechanism to assign importance weights $\lambda_1, ..., \lambda_M$ to each test case. Details of the selected test cases, their scenario features, and the corresponding weights are provided in the Supplementary Figs. 2, 6.

The accident rate for the SUT is then computed as follows:

$$\text{Accident} - \text{Rate} - \text{SUT} = \Gamma_1 \Gamma_2 \frac{\sum_{j=1}^{M} \lambda_j I_j}{\sum_{j=1}^{M} \lambda_j D_j} \quad (3)$$

where $j = 1, ...M$, and $\lambda_j$ as the weight for test case $j$

where $\Gamma_1$ compensates for the oversampling of traffic crashes during the construction of the test case pool, and $\Gamma_2$ accounts for the data processing ratio, as only 55,920 normal driving segments out of the 0.3 million cases are used for analysis.

Having calculated the accident rate for the SUT based on driving distance, we now introduce the calculation of Scaling Risk. To establish the human driver benchmark, which serves as the reference Accident-Rate-Human, we use the SHRP2 NDS dataset crash rate as a population-

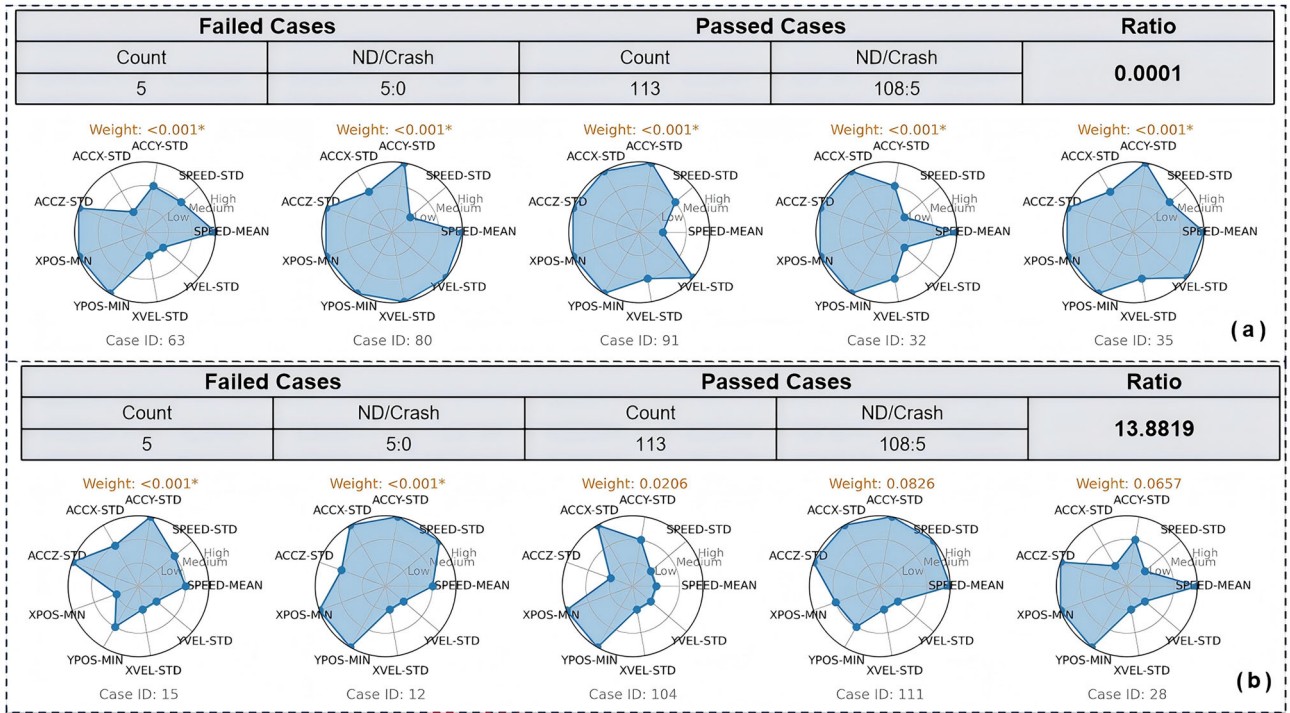

**Fig. 8 | Demonstration of two test trails for ADS safety validation over 118 sampling testable cases. a** Five failed cases with low weights leading to a safer outcome than human drivers. **b** Five failed cases with higher weights leading to a riskier outcome than human drivers. This demonstrates how specific test case ID failures result in different scaling risks compared to human drivers, depending on the associated weights.

level representation. In the SHRP2 NDS dataset, 90 crashes were recorded over approximately 33 million miles of driving. The crash rate for human drivers derived from the SHRP2 NDS is $1.695 \times 10^{-9}$ crashes/meters.

Since the testable cases are drawn from SHRP2 NDS real-world driving events, the safety level of the SUT is determined by evaluating its performance on the testable cases it passes or fails. The scaling risk (SR) in our study is defined as the ratio of the SUT's crash rate to the crash rate of benchmark human drivers, particularly in terms of being involved in severe traffic crashes. In other words, it is a relative risk measure that compares the SUT's safety performance with that of human drivers. The SR can be calculated as the follows:

$$SR = \frac{Accident - Rate - SUT}{Accident - Rate - Human} = \frac{Accident - Rate - SUT}{1.695 \times 10^{-9} \text{ crashes/meters}}$$

$$(4)$$

Figure 8 presents two demonstration trials, each accompanied by a radar plot that highlights the driving characteristics of the failed test cases. The overall risk assessment of the SUT can vary significantly depending on which specific test cases it fails. In both trials, the SUT fails exactly 5 cases while passing the remaining 113; however, the resulting risk assessments differ significantly. In Fig. 8b, if the SUT fails cases 15, 12, 104, 111, and 28, its risk level would be approximately 13.8 times higher than that of a human driver. The weights of the failed test cases play a critical role in determining the overall risk. Cases with higher weights have a more significant impact on risk assessment. For example, in the second trial, three of the five failed cases also have associated weights greater than 0.02, substantially contributing to the increased risk.

The characteristics of the failed test cases reflect the relevance of real-world traffic scenarios. For example, in Fig. 8b, the five failed cases show low values in *xvel−std* and *yvel−std*, indicating minimal interaction with surrounding vehicles. These types of scenarios are relatively common in real-world traffic; thus, failing to handle such cases adequately can significantly increase the overall risk of the SUT.

In contrast, in Fig. 8a, the five failed cases (ID: 63, 80, 91, 32, 35) result in safer performance compared to human drivers. All five of these cases are associated with weights smaller than 0.001. Notably, these cases share common characteristics: large values in *accz-std*, *xpos − min*, and *ypos − min*, suggesting a bumpy driving scenario with few nearby vehicles. Such conditions are rare in real-world settings, and their low weight reflects their limited impact in determining the overall safety risk of the SUT. The wide range of scaling risk values (0.0001 to 13.38 in Fig. 8) does not mean the ADS's risk literally fluctuates within this range during a single evaluation. Instead, it indicates that depending on which specific scenarios the ADS fails, the interpretation of the test results could be different. In this demonstration, the extreme values (0.0001 and 13.38) reflect the best and worst case combinations among the five selected failure cases.

We additionally derive SR's corresponding confidence interval to better quantify the uncertainty of SR. Specifically, the 95% confidence interval for the log value of the scaling risk can be constructed as follows:

$$\log SR \pm 1.96 \times \sqrt{\frac{\sum_{j=1}^{M} \lambda_j^2 r_j (1 - r_j)}{\left(\sum_{j=1}^{M} \lambda_j r_j\right)^2}}$$

$$(5)$$

where $r_j = (I_j + 0.5)/2$. If the lower bound of the established confidence interval is larger than 1, then we have 95% confidence to say the SUT could be riskier than human drivers. Detailed derivations on the confidence interval can be found in the supplementary material.

To validate the consistency of the proposed measure, we treat SHRP2 drivers as the SUT by letting $I_j$ represent crashes in each test case. Given the SHRP2 NDS baseline accident rate, the resulting ratio should be close to 1, and the value of 1 should be within the 95% confidence interval if our calculations are accurate. Using data from

the 118 test cases, along with their associated weights and distances, the calculated accident rate for the SHRP2-as-SUT scenario is $1.993 \times 10^{-9}$ crashes/meters. This results in an SR value of 1.176, the 95% confidence interval [0.441, 3.138], 95% confidence interval [0.441, 3.138], proving that our method is statistically validated. For Fig. 8b, out of the 118 selected cases, the point estimation of the scaling risk is 13.8819, the 95% confidence interval of [6.53, 29.50], indicating that, under this failure pattern, the SUT is significantly riskier than human drivers.

## Discussion

Motivated by the question of how we can effectively select a set of test cases to validate the safety functions of ADS? Given the complexity of real-world driving environments, this task becomes particularly difficult due to the large number of driving scenarios. To address this, we introduce the KTCS approach for selecting test cases from the largest-scale naturalistic driving study. This method strikes a balance between coverage and representativeness of the test case pool by integrating an attention mechanism with stochastic sampling. It helps to accelerate the validation process without compromising its reliability. This paper introduces a method that effectively represents the vast and complex real-world driving space with just a few dozen carefully selected scenarios. We demonstrate the use of the selected test cases with a scaling risk measure, which enables a direct safety performance comparison between ADS and human drivers, providing a theoretical guarantee for the validity of ADS safety evaluations.

Several potential extensions can be explored in future studies. Building on these insights, recent advances in deep generative modeling can be leveraged to create rare or unseen driving scenarios and to sample variations around the identified failure cases[34,35]. By amplifying these challenging situations, generative models can produce a dense cloud of near-miss trajectories, and testing the ADS against this expanded set yields stronger statistical evidence when comparing its safety performance to that of human drivers. Encoding these test cases in a standard format, such as OpenSCENARIO[36], will enable their integration into a growing, field-validated library that can be shared across industry and regulatory agencies. Insights gained from these evaluations can guide more rigorous simulations and field trials on industry-standard platforms[5], supporting scalable validation and targeted fine-tuning of ADS performance in controlled yet dynamic virtual environments or on dedicated test tracks. Researchers can also leverage the selected testable cases to inform the development of rigorous ADS safety test protocols, using industry-standard software or constructing field operation tests on dedicated test tracks.

The proposed KTCS algorithm is dataset-agnostic and can be applied to a wide range of naturalistic driving studies. We build test cases from strategically sampled subsets of the SHRP2 NDS. Future research could scale this methodology to the full SHRP2 database or apply it to other naturalistic driving studies and industry crowd-sourcing initiatives. Doing so would capture evolving traffic patterns and vehicle technologies, and refine human-performance baselines for modern ADS evaluations. These extensions could expand the pool of candidate test cases to hundreds of millions. At such a scale, standard kernel operations become computationally expensive. Recent advances in statistical computing offer viable solutions, such as low-rank approximations to compress kernel functions[37], or mini-batch kernel algorithms that partition data into blocks for parallel processing[38]. These strategies can make large-scale kernel analyses computationally feasible.

## Methods

With appropriate data pre-processing, let $x_1,..,x_N$ represent the test case pool and each data point $x_i \in \mathbb{R}^d$. We select a subset $z_1,..,z_M$ from the original set $x_1,..,x_N$. As we try to use as few samples as possible, trying to approximate the original large sample sizes, here we let $M \ll N$. Ideally, this subset $z_1,..,z_M$ contains both common cases and corner cases, as entire sets, the probability distribution function, constructed based on the selected subsets $z_1,..,z_M$, approximates the distribution of $x_1,..,x_N$.

Considering the potentially high dimensionality of the feature space ($d$) and the nonlinear dependencies among the features, we conduct our analysis in the reproducing kernel Hilbert space (RKHS)[24]. As a nonparametric framework, RKHS avoids the need to explicitly estimate the high-dimensional covariance matrix of the input features, a procedure that is often unstable and unreliable in such settings[39]. Beyond this practical advantage, RKHS is well-suited to capture nonlinear relationships and higher-order feature interactions, and has been successfully applied to data with complex covariance structures[40,41].

### Evaluation metrics

We use the kernel methods to measure the two objectives in this study: representativeness and coverage. Consider in space $\mathbb{X}$, defined $\mathcal{K}$ as a kernel function mapping $\mathbb{X} \times \mathbb{X} \to \mathbb{R}$. For example, the Radial basis function (RBF) kernel $\mathcal{K}(z_t, z_s) = \exp\left(\frac{-\|z_t - z_s\|_d^2}{2\sigma^2}\right)$ is a popular one. We define $\mu = \int \phi(x) d\mathbb{P}(x)$, as the kernel mean embedding at the population level. $\phi : \mathbb{X} \to \mathcal{H}$ represents a feature map transforming the $x_1, ..., x_N \in \mathbb{X}$ into Hilbert space. The feature map satisfies $\mathcal{K}(z_t, z_s) = \langle \phi(z_t), \phi(z_s) \rangle_{\mathcal{H}}$ and $\phi(z) = \mathcal{K}(z, \cdot)$.

Achieving good coverage performance requires comprehensively spanning the entire data space. Given the inherent complexity of high-dimensional space, we start our discussion by considering a one-dimensional space defined as $z_1, ..., z_M$. If the relative distances between adjacent points are large, the corresponding point configuration exhibits better space-filling properties, thereby providing more comprehensive coverage of the data space[31,42]. This relative distance among the $M$ points $z_1, ..., z_M$ can be expressed as follows:

$$\frac{1}{2M(M-1)} \sum_{i=1}^{M} \sum_{j \neq i}^{M} (z_i - z_j)^2 = \frac{1}{M-1} \sum_{i=1}^{M} (z_i - \bar{z})^2 = Variance \quad (6)$$

Thus, in a one-dimensional space, a higher variance among $z_1, ..., z_M$ indicates that these points are more widely dispersed, indicating a better coverage ability. Motivated by the discussion in Eq. (6), we consider using the variance as a measure of coverage within a multidimensional space in the RKHS. Based on the properties of reproducing kernel Hilbert space (RKHS),[43] investigate the variance of multidimensional data. In our case, based on the sub-selected data, $z_1, .., z_M$, where $z_i \in \mathbb{R}^d$, and $\| . \|_{\mathcal{H}}$ denotes the norm under Hilbert space. We denote $\nu_z^2 = \mathbb{E} z \sim \mathbb{P} \| \mathcal{K}(z, \cdot) - \mu_{\mathbb{P}} \|_{\mathcal{H}}^2$ as the variance for the multivariate $z$ in the RKHS, which can be decomposed through the following term:

$$\nu_z = \mathbb{E}_{z \sim \mathbb{P}}[\mathcal{K}(z, z)] - \underbrace{\mathbb{E}_{z, z' \sim \mathbb{P}}[\mathcal{K}(z, z')]}_{InformationPotential(IP)} \quad (7)$$

The kernel function $\mathcal{K}(z, z')$ measures the distance between cases $z$ and $z'$ by evaluating all the feature coordinates simultaneously. This allows pairwise similarities to embed second-order dependencies among the features in the RKHS[44]. By aggregating these kernel similarities, the Information Potential (IP) approach implicitly incorporates moments of all orders, such as means, variances, and covariances. This ensures that coverage is assessed on the entire joint distribution of the features, rather than just the marginal distributions[23].

If we let the kernel function maintain the property that $\mathcal{K}(z, z') \leq \mathcal{K}(z, z) = K$, it is easy to verify that the first component is $\mathbb{E}_{z \sim \mathbb{P}}[\mathcal{K}(z, z)] = K$. For example, if we are using the RBF kernel as

**BOX 1**

# Algorithm for stochastic samplings through optimizations

- Do the optimization $\theta_{\text{opt}} = \text{argmin}\theta \sum_{i=1}^{N} \sum_{j \neq i}^{N} w_\theta(\boldsymbol{x}_i) w_\theta(\boldsymbol{x}_j) \mathcal{K}(\boldsymbol{x}_i, \boldsymbol{x}_j)$

- Calculate the importance measures as $w_{\theta_{\text{opt}}}(\boldsymbol{x}_1), .., w_{\theta_{\text{opt}}}(\boldsymbol{x}_N)$

- Pareto order sampling[46] is implemented:

  - Generate $U_1, \ldots U_N \sim \text{Uniform}(0,1)$

  - Calculate $Q_i = \frac{U_i}{1-U_i} \times \frac{1 - w_{\theta_{\text{opt}}}(\boldsymbol{x}_i)}{w_{\theta_{\text{opt}}}(\boldsymbol{x}_i)}$ for $i = 1, \ldots, N$

  - Rank $Q_1, \ldots, Q_N$, we select the $M$ smallest ones.

  - The corresponding cases labeled as $\boldsymbol{z}_1, \ldots \boldsymbol{z}_M$

- Output: $\boldsymbol{z}_1, \ldots \boldsymbol{z}_M$ as the selected points through stochastic sampling.

$\mathcal{K}(\boldsymbol{z}_t, \boldsymbol{z}_s) = \exp\left(\frac{-\|(\boldsymbol{z}_t - \boldsymbol{z}_s\|_d^2}{2\sigma^2}\right)$, then $\mathbb{E}_{\boldsymbol{z} \sim \mathbb{P}}[\mathcal{K}(\boldsymbol{z}, \boldsymbol{z})] = 1$. It is obvious then that, as the first term $\mathbb{E}_{\boldsymbol{z} \sim \mathbb{P}}[\mathcal{K}(\boldsymbol{z}, \boldsymbol{z})]$ is a constant, the dominant major term in $v_{\boldsymbol{z}}$ will be the second term, and we define it to be Information Potential (IP) in Eq. (7). A similar technique has also been used in[23]. With the sample data $\boldsymbol{z}_1, ...\boldsymbol{z}_M$ where $\boldsymbol{z}_i \in \mathbb{R}^d$, we can estimate the IP for multivariate $\boldsymbol{z}$ based on the properties of RKHS as:

$$IP_{\boldsymbol{z}} = \frac{1}{M(M-1)} \sum_{t=1}^{M} \sum_{s=1, s \neq t}^{M} \mathcal{K}(\boldsymbol{z}_t, \boldsymbol{z}_s) \qquad (8)$$

The minimization of the information potential corresponds to maximizing the variance of the feature vectors, which increases the coverage of data samples. In our situation, selecting the corner cases can be represented by minimizing the IP functions, which pushes more diversification in selecting the instances.

Secondly, we utilize the MMD as the statistical criterion to measure representativeness. MMD is an important statistical metric to quantify the statistical distance between two distributions in the RKHS[24]. In our study, we aim to use the selected cases $\boldsymbol{z}_1, ...,\boldsymbol{z}_M$ to approximate the original data sets $\boldsymbol{x}_1, ...,\boldsymbol{x}_N$. Here, we use the term $\mathbb{P}_{\boldsymbol{x}}$ to define the probability distribution constructed based on the points $\boldsymbol{x}_1, ...,\boldsymbol{x}_N$, and we define $\mathbb{P}_{\boldsymbol{z}}$ as the probability distribution constructed with $\boldsymbol{z}_1, ...,\boldsymbol{z}_M$. Based on the sample data,[24] use an approach of U statistic to estimate the squared MMD as follows:

$$MMD_u^2(\mathbb{P}_{\boldsymbol{x}}, \mathbb{P}_{\boldsymbol{z}}) = \frac{1}{N^2} \sum_{i=1}^{N} \sum_{j=1}^{N} \mathcal{K}(\boldsymbol{x}_i, \boldsymbol{x}_j) + \frac{1}{M^2} \sum_{i=1}^{M} \sum_{j=1}^{M} \mathcal{K}(\boldsymbol{z}_i, \boldsymbol{z}_j) - \frac{2}{MN} \sum_{i=1}^{N} \sum_{j=1}^{M} \mathcal{K}(\boldsymbol{x}_i, \boldsymbol{z}_j)$$

$$(9)$$

Smaller values of $MMD_u^2(\mathbb{P}_{\boldsymbol{x}}, \mathbb{P}_{\boldsymbol{z}})$ indicate xthat the selected cases $\boldsymbol{z}_1, ...,\boldsymbol{z}_M$ closely approximate the target distribution $\mathbb{P}_{\boldsymbol{x}}$.[24] theoretically discuss the asymptotic properties of $MMD_u^2(\mathbb{P}_{\boldsymbol{x}}, \mathbb{P}_{\boldsymbol{z}})$ toward $MMD^2(\mathbb{P}_{\boldsymbol{x}}, \mathbb{P}_{\boldsymbol{z}})$ based on the properties of U statistics and RKHS. Based on these established theoretical guarantees, we will use this $MMD_u^2(\mathbb{P}_{\boldsymbol{x}}, \mathbb{P}_{\boldsymbol{z}})$ to quantify the representativeness of selected cases $\boldsymbol{z}_1, ...,\boldsymbol{z}_M$.

**Stochastic sampling algorithm**

One way to generate stochastic samplings from the candidate test case pool $\boldsymbol{x}_1, ...,\boldsymbol{x}_N$ is to sample $M$ points based on a sequence of assigned importance measures $w_1, ...,w_N$. In this context, larger values of the importance measure $w_i$ for the case $\boldsymbol{x}_i$ indicate a higher likelihood of inclusion in the subsampled $M$ cases. To ensure the preservation of probabilistic interpretation, we enforce the constraints

that the assigned importance measures $w_j > 0$ for all $j = 1,..., N$ and $\sum_{i=1}^{N} w_i = 1$.

The calculation of IP in Eq. (8) treats each sample in $\boldsymbol{x}_1, ...,\boldsymbol{x}_N$ with an equal importance measure, i.e., each of the cases in the test case pool has the importance measure as $1/N$. Here, we consider $\boldsymbol{x}_1, ...,\boldsymbol{x}_N$ to be associated with unequally important measures. The points that accompany more important measures should have higher weights in the estimation of IP, and vice versa. In this context, we define the IP with importance measures for the associated points $w_1, ...w_N$ as $IP_{\boldsymbol{x}, \text{wgt}}$ in the following:

$$IP_{\boldsymbol{x}, \text{wgt}} = \sum_{t=1}^{N} \sum_{s \neq t, s=1}^{N} w_t w_s \mathcal{K}(\boldsymbol{x}_t, \boldsymbol{x}_s) \qquad (10)$$

Based on Eq. (10), the key is to find optimal values in the importance measures that can minimize IP. Then, such optimal importance measures could be used to generate $M$ sub-samples. With the importance measures $w_1, ...w_N$, the optimization implemented in Step 1 is equivalent to solving $minIP_{\boldsymbol{x}, \text{wgt}}$. For this optimization problem, we denote a vector $\mathbf{w} = (w_1, ...,w_N)$, and the matrix as $\mathcal{K}_{XX}$ whose $(i, j)$ th entry is $(\mathcal{K}_{XX})_{ij} = \mathcal{K}(\boldsymbol{x}_i, \boldsymbol{x}_j)$. Thus, we can reformulate $IP_{\boldsymbol{x}, \text{wgt}}$ as $IP_{\boldsymbol{x}, \text{wgt}} = \mathbf{w}^\top \mathcal{K}_{XX} \mathbf{w}$. Mathematically, minimizing the information potential with importance measures is as:

$$\min_{\mathbf{w}} IP_{\boldsymbol{x}, \text{wgt}} = \min_{\mathbf{w}} \mathbf{w}^\top \mathcal{K}_{XX} \mathbf{w} \ s.t. \ \|\mathbf{w}\|_1 = 1 \qquad (11)$$

We can easily derive a closed-form solution for this optimization problem and the optimal sampling weight $\mathbf{w}^*$ calculated as $\mathbf{w}^* = \frac{\mathcal{K}_{XX}^{-1} \mathbf{1}_N}{\mathbf{1}_N^\top \mathcal{K}_{XX}^{-1} \mathbf{1}_N}$, where $\mathbf{1}_N^\top$ denotes an $N$-length all ones vector. Unfortunately, calculating the exact / pseudo inverse matrix of $\mathcal{K}_{XX}$ is neither computationally efficient nor accurate when the size $N$ becomes sufficiently large[45]. To this end, we propose a more efficient approach to approximate the optimization problem above. We set the importance measure for case $\boldsymbol{x}_i$ as $w_i$, which results in a weighted set $\{(\boldsymbol{x}_1, w_1),...,(\boldsymbol{x}_N, w_N)\}$. Larger values in $w_i$ indicate a higher chance of the corresponding case $\boldsymbol{x}_i$ being among the $M$ selected cases. Rather than treating $w_1, ...w_N$ as independent free parameters to be optimized, which would be computationally expensive, we approximate their optimal values using a shared parametric function. That is, we set the importance measure for the case $\boldsymbol{x}_i$ as $w_\theta(\boldsymbol{x}_i)$, where the function is parameterized by $\theta$ as $w_\theta : \mathbb{R}^d \to \mathbb{R}$. Specifically, we set this function $w_\theta(\cdot)$ as a simple multiple-layer perceptron (one layer in our studies proves very effective), with Softmax as the activation function. The parameter $\theta$ should

fall on the $p$-dimensional unit ball, i.e., $\theta \in \mathcal{B}^p$. With these, we define $\mathcal{E}_X(\theta)$ as follows:

$$\mathcal{E}_X(\theta) = \sum_{i=1}^{N} \sum_{\substack{j=1 \\ j \neq i}}^{N} w_\theta(\boldsymbol{x}_i) w_\theta(\boldsymbol{x}_j) \mathcal{K}(\boldsymbol{x}_i, \boldsymbol{x}_j) \quad (12)$$

In order to select the $M$ sub-samples with the minimal IP, we first minimize $\mathcal{E}_X(\theta)$, so as to approximate the optimal importance measures which lead to the minimum in the IP. We can approximate the optimal importance measures through $w_{\theta_{\mathrm{opt}}}(\boldsymbol{x}_1), \dots w_{\theta_{\mathrm{opt}}}(\boldsymbol{x}_N)$, where the parameters $\theta_{\mathrm{opt}}$ can be achieved through the following optimization process:

$$\theta_{\mathrm{opt}} = \mathrm{argmin}_{\theta \in \mathcal{B}^p} \, \mathcal{E}_X(\theta) = \mathrm{argmin}_{\theta \in \mathcal{B}^p} \sum_{i=1}^{N} \sum_{\substack{j=1 \\ j \neq i}}^{N} w_\theta(\boldsymbol{x}_i) w_\theta(\boldsymbol{x}_j) \mathcal{K}(\boldsymbol{x}_i, \boldsymbol{x}_j) \quad (13)$$

Based on the optimized function $w_{\theta_{\mathrm{opt}}}(\cdot)$ for $\boldsymbol{x}_1, \dots, \boldsymbol{x}_N$, we generate importance measures $w_{\theta_{\mathrm{opt}}}(\boldsymbol{x}_1), \dots, w_{\theta_{\mathrm{opt}}}(\boldsymbol{x}_N)$. Pareto order sampling is then applied to generate exactly $M$ sub-samples $\boldsymbol{z}_1, \dots, \boldsymbol{z}_M$ based on these importance measures. A more detailed discussion of Pareto order sampling can be found in[46]. We first generate random variables $U_1, \dots, U_N$ from a Uniform(0,1) distribution. Then, for each case $i$, we compute the Pareto ranking variables as $Q_i = \frac{U_i}{1 - U_i} \times \frac{1 - w_{\theta_{\mathrm{opt}}}(\boldsymbol{x}_1)}{w_{\theta_{\mathrm{opt}}}(\boldsymbol{x}_i)}$ for $i = 1, \dots, N$. Among all $Q_1, \dots, Q_N$, we select the $M$ smallest values. The corresponding $M$ samples are then selected as the final sub-sample, weighted according to their importance measures $w_{\theta_{\mathrm{opt}}}(\boldsymbol{x}_1), \dots, w_{\theta_{\mathrm{opt}}}(\boldsymbol{x}_N)$. We summarize the stochastic sampling procedure in Box 1.

## Attention-based mechanism

Based on the selected points, we next investigate how to assign distribution alignment weights $\lambda_1, \dots, \lambda_M$ to each of the selected points $\boldsymbol{z}_1, \dots, \boldsymbol{z}_M$ so as to increase representativeness. The objective in optimizing the weights is to minimize the MMD between the weighted version of $\boldsymbol{z}_1, \dots, \boldsymbol{z}_M$ and the original dataset $\boldsymbol{x}_1, \dots, \boldsymbol{x}_N$. Here, under this setup, we can estimate the MMD between $\mathbb{P}_{\boldsymbol{x}}$ and the $\mathbb{P}_{\boldsymbol{z}^{\cdot}}$ as follows:

$$MMD_u^2(\mathbb{P}_{\boldsymbol{x}}, \mathbb{P}_{\boldsymbol{z}^{\cdot}}) = \frac{1}{N^2} \sum_{i=1}^{N} \sum_{j=1}^{N} \mathcal{K}(\boldsymbol{x}_i, \boldsymbol{x}_j) + \sum_{i=1}^{M} \sum_{j=1}^{M} \lambda_i \lambda_j \mathcal{K}(\boldsymbol{z}_i, \boldsymbol{z}_j) - 2 \frac{1}{N} \sum_{i=1}^{N} \sum_{j=1}^{M} \lambda_j \mathcal{K}(\boldsymbol{x}_i, \boldsymbol{z}_j) \quad (14)$$

As the $\frac{1}{N^2} \sum_{i=1}^{N} \sum_{j=1}^{N} \mathcal{K}(\boldsymbol{x}_i, \boldsymbol{x}_j)$ is a constant as long as the original dataset is fixed, so as optimizing the $MMD_u^2(\mathbb{P}_{\boldsymbol{x}}, \mathbb{P}_{\boldsymbol{z}^{\cdot}})$ is equivalent with optimizing the remaining terms. We use a vector $\boldsymbol{\lambda}$ to represent $\boldsymbol{\lambda} = (\lambda_1, \dots, \lambda_M)^\top$ as a $M \times 1$ vector, and for simplicity, we define $\overline{\mathbf{K}} = \frac{1}{N} \mathbf{1}_N^\top \mathcal{K}_{\boldsymbol{xz}}$ as an $M \times 1$ vector, so as minimizing the $MMD_u^2(\mathbb{P}_{\boldsymbol{x}}, \mathbb{P}_{\boldsymbol{z}^{\cdot}})$ is equivalent as:

$$\text{minimize}_{\boldsymbol{\lambda}} \, \frac{1}{2} \boldsymbol{\lambda}^\top \mathcal{K}_{\boldsymbol{zz}} \boldsymbol{\lambda} - \boldsymbol{\lambda}^\top \overline{\mathbf{K}} \quad \text{s.t.} \sum_{i=1}^{M} \lambda_i = 1 \text{ and } \lambda_1, \dots, \lambda_M > 0 \quad (15)$$

Thus, we can achieve the selected points $\boldsymbol{z}_1, \dots, \boldsymbol{z}_M$ and associated weights as $\lambda_1, \dots, \lambda_M$. These $\lambda$-weights serve as the distribution alignment weights, used to realign the empirical distribution of the subset $\boldsymbol{z}_1, \dots, \boldsymbol{z}_M$ with that of the entire candidate test case pool $\boldsymbol{x}_1, \dots, \boldsymbol{x}_N$.

## KTCS algorithm

The KTCS algorithm, designed to simultaneously achieve a balance between representativeness and coverage. The logic of the KTCS

method is to construct a weighted subset of data that allows for closely approximates the inference using the entire test case pool. Coverage ensures that the selected data points encompass a diverse range of scenarios, while representativeness ensures the selected subset closely approximates the behavior of the entire dataset. Achieving optimal coverage and representativeness is equivalent to minimizing IP and MMD simultaneously, as smaller values in IP and MMD indicate greater diversity and better representativeness. We state our optimization objective as follows:

$$\text{Step 1}: \underset{\boldsymbol{z}_1, \dots \boldsymbol{z}_M}{\mathrm{argmin}} \, IP_{\boldsymbol{z}} = \underset{\boldsymbol{z}_1, \dots \boldsymbol{z}_M}{\mathrm{argmin}} \, \frac{1}{M(M-1)} \sum_{t=1}^{M} \sum_{\substack{s=1, s \neq t}}^{M} \mathcal{K}(\boldsymbol{z}_t, \boldsymbol{z}_s) \text{ where } \boldsymbol{z}_1, \dots \boldsymbol{z}_M \in \{\boldsymbol{x}_1, \dots, \boldsymbol{x}_N\} \quad (16)$$

$$\text{Step 2}: \underset{\lambda_1, \dots \lambda_M}{\mathrm{argmin}} \, MMD_u^2(\mathbb{P}_{\boldsymbol{x}}, \mathbb{P}_{\boldsymbol{z}^{\cdot}}) = \underset{\boldsymbol{\lambda}}{\text{minimize}} \, \frac{1}{2} \boldsymbol{\lambda}^\top \mathcal{K}_{\boldsymbol{zz}} \boldsymbol{\lambda} - \boldsymbol{\lambda}^\top \overline{\mathbf{K}} \text{ s.t.} \sum_{i=1}^{M} \lambda_i = 1 \text{ and } \lambda_1, \dots, \lambda_M > 0 \quad (17)$$

where $\mathbb{P}_{\boldsymbol{z}^{\cdot}}$ means that the distribution is established by the points $\boldsymbol{z}_1, \dots, \boldsymbol{z}_M$ with associated weights $\lambda_1, \dots \lambda_M$. The kernel mean embedding for $\mathbb{P}_{\boldsymbol{z}^{\cdot}}$ is $\mu_{\mathbb{P}_{\boldsymbol{z}^{\cdot}}} = \sum_{j=1}^{M} \lambda_j \phi(\boldsymbol{z})$.

KTCS algorithm has two steps. Step 1 focuses on a discrete design, where the key idea is to select $M$ cases from the original pool of $N$ candidate test cases. The optimization variables are $w_1, \dots, w_N$ for $\boldsymbol{x}_1, \dots, \boldsymbol{x}_N$, and the objective is to minimize the IP. Minimizing IP promotes well-dispersed selections in the feature space, serving as a surrogate for diversity and coverage. In Step 2, we re-weight the $M$ selected cases so that their empirical distribution aligns as closely as possible with that of the full candidate pool. The optimization variables are $\lambda_1, \dots, \lambda_M$ for $\boldsymbol{z}_1, \dots, \boldsymbol{z}_N$, and the objective is to minimize MMD in Eq. (17), which quantifies representativeness. Together, this dual-step process creates a computationally efficient framework that selects and weights data points to maintain both diversity and representativeness, enabling accurate and scalable inference. Extensive Monte Carlo simulations to validate the performance of the KTCS algorithm are shown in the supplementary part.

## Theoretical discussion

This section presents the theoretical guarantees of the KTCS method. We first examine its ability to ensure scenario coverage. In particular, we analyze the theoretical behavior of the proposed approach by investigating the asymptotic properties of the IP metric, which quantifies coverage, under stochastic sampling of selected scenarios. Theorem 1 shows that when $N$ is sufficiently large, $IP_{\boldsymbol{z}}$ approximates well to $\mathcal{E}_X(\theta_{\mathrm{opt}})$, that is, the minimize of the information potential with weighted importance measures.

**Theorem 1**. Define $IP_{\boldsymbol{z}}$ as the information potential with the selected points through Pareto order sampling. When the size $N$ goes sufficiently large, the following holds:

$$IP_{\boldsymbol{z}} \to \mathcal{E}_X(\theta_{\mathrm{opt}}) \text{ as } N \to \infty \text{ where } \mathcal{E}_X(\theta_{\mathrm{opt}}) = \sum_{i=1}^{N} \sum_{\substack{j=1 \\ j \neq i}}^{N} w_{\theta_{\mathrm{opt}}}(\boldsymbol{x}_i) w_{\theta_{\mathrm{opt}}}(\boldsymbol{x}_j) \mathcal{K}(\boldsymbol{x}_i, \boldsymbol{x}_j) \quad (18)$$

We next establish the asymptotic behaviors of $\mathcal{E}_X(\theta_{\mathrm{opt}})$ towards global minimization in the Information Potential defined as $\mathcal{E}(\theta^*)$ in the following theorem. Based on the discussions in Theorem 2, we find that the minimize of the IP with weighted importance measures $\mathcal{E}_X(\theta_{\mathrm{opt}})$ will finally converge towards global minimization in the IP defined as $\mathcal{E}(\theta^*)$ when the size $N$ goes sufficiently large,

**Theorem 2**. Assume that the parameter $\theta$ falls on the $p$ dimensional unit ball (i.e. $\theta \in \mathcal{B}^p$) $\epsilon$ is some constant, and $w_\theta(\boldsymbol{x}) \leq s$. With at least a

probability of $1 - \delta$:

$$\mathcal{E}(\theta^*) - \mathcal{E}_X(\theta_{\text{opt}}) \leq R_N + \frac{s^2 K}{\sqrt{N}} \sqrt{2 \log(2/\delta) + 2p \log\left(1 + \frac{2}{\epsilon}\right)} \quad (19)$$

$\mathbb{X}\mathcal{K}(\boldsymbol{x}, \boldsymbol{x}')w_\theta(\boldsymbol{x})w_\theta(\boldsymbol{x}')d\mathbb{P}(\boldsymbol{x}, \boldsymbol{x}')$ and $R_N$ is the empirical Rademacher complexity.

Theorem 1 and Theorem 2 show that the KTCS can select test cases with optimized coverage when the number of instances within the test case pool goes sufficiently large, as the IP of the subselected cases will finally converge to the global minimization in the IP, achieving the goal of Step 1 in Eq. (16).

Next, we theoretically investigate the ability of KTCS to ensure representativeness. We investigate the theoretical properties of the proposed method, KTCS. We define $\mu = \int \phi(\boldsymbol{x})d\mathbb{P}(\boldsymbol{x})$ as the population level kernel mean embedding. Our estimator, based on the selected samples with optimized weights, is constructed as $\widehat{\mu}_M = \sum_{j=1}^{M} \lambda_j \phi(\boldsymbol{z}_j)$. The following theorem states the error bound from our proposed estimator and the population level $\mu$. Theorem 3 shows that $N$ goes sufficiently large, $\widehat{\mu}_M$ approximates $\mu$.

**Theorem 3.** Assumes the kernel function satisfying $\mathcal{K}(\boldsymbol{z}, \boldsymbol{z}') \leq \mathcal{K}(\boldsymbol{z}, \boldsymbol{z}) = K$ where $\boldsymbol{z} \neq \boldsymbol{z}'$. Define $\delta, c, c', \lambda$ as some constants and $\delta \in (0, 1)$, $\mathcal{N}_\infty(\lambda)$ as some constant depending on $\lambda$. Then with at least a probability of $1 - \delta$, the following bound holds:

$$\|\widehat{\mu}_M - \mu\|_{\mathcal{H}} \leq \frac{2K\sqrt{2\log(6/\delta)}}{\sqrt{N}} + c'K\frac{\sqrt{2\log(6/\delta)}}{\sqrt{M}}$$
$$+ \sqrt{3\lambda} \times \left( \frac{4\sqrt{\mathcal{N}_\infty(\lambda)}\log(12/\delta)}{M} + \sqrt{\frac{2c\nu_z \log(12/\delta)}{M}} \right) \quad (20)$$

where $\nu_{\boldsymbol{z}}^2 = \mathbb{E}(\| \phi(\boldsymbol{z}_j) - \mathbb{E}(\phi(\boldsymbol{z}_j)\|_{\mathcal{H}}^2)$ is established on the selected points.

**Remark 1.** If set $M = \mathcal{O}(N^{1/2})$, under some mild conditions discussed in the supplementary part, then $\|\widehat{\mu}_M - \mu\|_{\mathcal{H}} = \mathcal{O}(\frac{1}{\sqrt{N}})$.

**Remark:** Theorem 3 shows that, when setting the number of selected cases to $M = \mathcal{O}(N^{1/2})$, then the KTCS method can achieve sufficient representativeness because the constructed density distribution using the selected cases can approximate the original test case pool. All technical proofs are presented in the supplementary part.

## Data availability

The scenario feature data and non-PII Strategic Highway Research Program Naturalistic Driving Study data are available under restricted access. Researchers may obtain access by submitting a Data Use License application, which requires specification of the requested data, documentation of Institutional Review Board (IRB) approval, institutional authorization, and a data security plan. Detailed instructions for data access are available at the following links: https://www.trb.org/StrategicHighwayResearchProgram2SHRP2/ SHRP2DataSafetyAccess.aspxhttps://insight.shrp2nds.us/. The data underlying the figures presented in this paper are provided in the accompanying Source Data file. Source data are provided with this paper.

## Code availability

The source code and implementation details are publicly available at Code Ocean through https://doi.org/10.24433/CO.9203840.v1

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

## Acknowledgments

We thank Drs. Miguel Perez, Jon Hankey, Kevin Kefauver and Zac Doerzaph for their valuable guidance on naturalistic driving study data and automated driving systems testing and validation.

## Author contributions

F. Guo. conceptualized the study and provided the data. C. Qian, J.Xu. and F. Guo developed the methodology. X. Xing validated the findings. C. Qian. and J.Xu performed the formal analysis. C. Qian. wrote the original draft, while all authors revising the editing the manuscript. F. Guo and X. Xing provided supervision.

## Competing interests
The authors declare no competing interests.
