## [Transparent Peer Review file · Nature Communications]

Test Case Sampling Optimization for Safety Validation of Automated Driving Systems

Corresponding Author: Professor Feng Guo

Version 0:

Reviewer comments:

Reviewer #1

(Remarks to the Author)

The article proposes a novel approach for selecting test cases from naturalistic driving studies for safety validation of automated driving system. The approach relies on the kernel embedding (of distributions) method and is based on an algorithm for test case selection that tries to achieve both coverage and representativeness of the multivariate distribution density of the driving features. The approach is then applied to 118 cases sampled from the Second Strategic Highway Research Program.

MAJOR POINTS

In statistics, any inferential problem aims to analyse a sample taken from a population in order to answer relevant questions about the population itself. In this article, much effort is put into obtaining a sample that can represent another sample (which is itself a sample of a larger sample). Specifically, the 55,920 driving segments are a sample of the original 300,000 driving segments, which are a sample of SHRP2, which is a sample of the population of all cars in circulation in the US. Proving that the final 118 selected cases are a good approximation of the 55,920 driving samples does not prove that these cases are representative of the population. At the very least, it would be important to quantify the uncertainty of all results, bearing in mind that these 300,000 segments are neither the full SHRP2 dataset nor the population.

Equation (5) introduces the variance in a one dimensional space and its role to achieve coverage is subsequently discussed. Then the same concept is extended to the multivariate case. When dealing with multivariate distributions, however, covariances become important. These are completely ignored in the definition of information potential in equations (6) and (7). Ignoring covariances likely implies that the coverage is reached in a sub-optimal way.

Page 11: The definition of the weights w_i used in equation 11 is rather obscure and needs to be clarified. As I understand it, the weights are a function of the observed features used to minimise the information potential. What is the implication of this choice?

The KTCS algorithm is based on two steps. The first step tries to achieve coverage (maximizing the variance) while the second step tries to achieve representativeness by minimizing the distance between the distribution of the smaller sample with the distribution of the larger sample. The two steps are not independent because a distribution is "defined" by its moments, including its variance. What I see in this case, is a first step where M cases are selected (in a sub-optimal way because covariances are ignored) and then some weights (the lambdas) which are supposed to "reflecting the importance of each scenario in driving" (end of page 2). What I see is that the lambdas simply try to adjust what is done in step 1, where the actual "shape" of the distribution density in the high dimensional space is ignored because the variance on its own cannot describe it. Also, I do not see how these weights are related to the importance of the driving scenarios. The same weights are used in Equation (3) to define the accident rate so they have an important role, but I do not see a validation of this weights and their meaning (also because they have been computed on only 118 cases). I also suspect that those weights are "responsible" for the problem discussed in my next point.

When it comes to apply the methodology to quantify the safety level of an ADS the results do not seem to be satisfactory.

The scaling risk in Figure 7 goes from 0.0001 to 13.8819 depending on which the 5 failed cases are. How is your approach useful in practice if the scaling risk may change by a factor of 138,800 by changing the trial? I believe the case study is too weak, and/or it shows that 118 cases are not enough for the safety validation of the driving system (which is the claimed goal of the approach).

MINOR POINTS

Page 1: "As shown in Figure 1(a), these assumptions can diverge significantly from actual on-road conditions". Figure 1(a) only shows some invented examples which (I guess) are not taken from the SHRP2 dataset.

Page 1: The Second Strategic Highway Research Program is introduced without a reference.

Page 2: I believe "Too much focus on common cases" and "Too much focus on corner cases" are swapped in Figure 1(c).

Page 2: "Representativeness ensures that the cases collectively approximate the distribution of real-world scenarios". At this point in the article, the distribution (of driving characteristics) has not yet been defined, so it is not clear which distribution it is.

Page 2: "a small number of strategically selected green points around the core of the testable space can effectively represent complex driving scenarios". Complex in terms of what? I do not see why the cases around the core necessarily represent complex scenarios.

Page 2: "Coverage ensures that cases include diverse driving scenarios, particularly in low-density areas". Similarly, density has not yet been formally introduced.

Page 2: "can effectively target high-risk areas but may lack fidelity". What is fidelity? It is not formally defined in the article.

Page 2: "To achieve this, an attention mechanism is introduced that assigns weights to the selected cases". A reference about the attention mechanism is needed.

Page 2: "Higher weights indicate more common scenarios, while lower weights correspond to less frequent ones". This interpretation of the weights is different from the interpretation elsewhere in the article, where a higher weight is associated with cases of higher importance. But being an important case is not related to being a frequent or less frequent case.

Page 3 first line: "performance based on the success or failure of each case". What is a success and what is a failure?

Page 3 first line: "Scaling risk reflects how much more risk the ADS carries compared to human drivers". Risk usually has a formal definition, depending on the context. What is risk in this case?

Page 3: "The proposed algorithm addresses the NP-hard problem of minimizing the information potential of driving data by assigning varying weights to selected test cases, thus ensuring comprehensive coverage of long-tailed critical driving scenarios." What is a tail in a multidimensional space?

Page 3: "To our knowledge, this is the first method to effectively use a small number of test cases to represent the entire real-world driving space while also providing theoretical guarantees for ADS safety validation." I do not think this can be claimed, for the reason explained in my first major point.

Page 3: "We employed importance sampling to construct a test case pool from traffic crashes". Written this way, it looks like the pool only contains crashes.

Page 3: "The test cases are categorized into two groups: normal driving and safety-critical events." What is normal driving? And what are safety-critical events? Just crashes or something else?

Page 4: "In total, we randomly selected about 300,000". What is the size of the full original pool?

Page 4: "We set $M = 0.5\sqrt{N}$, resulting in 118 selected cases for our analysis". Why this choice of M? And how do you know that the sample of M cases will include crashes? "In total, 55,920 normal driving segments and 90 crash events are used for the analysis", so the crashes are very small in number so I guess it is possible that the 118 cases do not include crashes(?)

Page 4: "The density distributions are generated using kernel density estimation, with darker blue indicating higher density regions." What about the white area? Figure 3 must include a color scale of the density.

Figure 3: Why are the selected cases (red markers) only shown for the novel approach?

Page 5: "MMD-Critic, on the other hand, focuses on a few long-tailed cases but lacks the ability to comprehensively represent safety-critical situations". How can you tell this? What is a safety critical situation in terms of driving features?

Page 5: "While MED demonstrates some capacity to cover long-tailed cases, our method selects more evenly distributed

points, particularly along boundary lines." What is a boundary line? In any case I do not see this as a good property. This is the result of maximizing the variances without considering the covariances.

Page 6: "Corner cases, characterized by long-tailed characteristics". Again, it is not clear what is a tail in a multidimensional space.

Figure 6(b) and (c). Are these percentages a quantile in multidimensional space or something else? This is not clear from either the text or the figure.

Page 9: "If the relative distance between any two adjacent points are far away each other, then they can have higher chance to cover the entire data space". What does it mean a higher chance? The data space limits are known because they are defined by the larger pool. The chance to cover the entire data space is deterministic.

Equation 5: the double sum is not equal to the variance. There is a 2x factor.

Page 10: "Smaller values of MMD2 can indicate the selected cases z_1, \dots, z_M closely approximate the distribution". What does it mean that they can? That they also cannot?

Page 10: "With the importance measures w_1, \dots, w_N , the optimization implemented in step 1 from Equation 15 should be equivalent as $\min \sum w_i \cdot \text{wt}_i$ ". "Should be equivalent" gives the impression that the authors are not sure.

Regards,

Prof. Francesco Finazzi

(Remarks on code availability)

The code only seems to reproduce figures and tables, but I could not find the algorithms described in the paper, in particular the KTCS algorithm.

Reviewer #2

(Remarks to the Author)

Ensuring the safety of Automated Driving Systems (ADS) requires a thorough and standardized approach to selecting test cases that encompass both common (representative) and rare (high-risk) driving scenarios. The authors identify several key challenges. First, the complexity of real-world driving makes it difficult to design comprehensive test scenarios, since real driving involves numerous factors such as vehicle dynamics, road conditions, and interactions with other vehicles. Second, the rarity of safety-critical events—accidents and near-crashes—means that vast amounts of driving data would be necessary to capture them if one relied solely on random observation in normal conditions.

To address these challenges, the paper leverages data from the Second Strategic Highway Research Program (SHRP2)—the largest-scale naturalistic driving study (NDS) in the United States. From this database, the authors aim to systematically select a small but strategically chosen set of test cases that reflects the distribution of real-world driving while also including more extreme corner cases.

They introduce Kernel Test Case Sampling (KTCS), a two-step algorithm to select a manageable subset of test cases from the large SHRP2 pool. The algorithm seeks to balance coverage (ensuring that sparse or corner-case scenarios are included) with representativeness (ensuring that the selected subset mirrors the overall driving distribution). Since the selection process draws on naturalistic driving data, the resulting test cases reflect actual behaviors, speeds, distances, and accident rates.

Additionally, the paper proposes a novel safety measure called Scaling Risk, which compares an ADS's crash rate directly to that of human drivers in SHRP2. This comparison helps close a gap in existing testing methods: government and industry generally benchmark ADS safety against typical human crash rates. The authors also provide theoretical analysis showing that, given a sufficiently large data pool, the weighted subset converges to an accurate representation of the underlying distribution. This "asymptotic" justification suggests that a small, well-chosen sample can remain statistically valid.

Despite these contributions, some limitations remain. The authors point out that using generative models to create additional corner cases not present in the NDS data could help cover extremely rare situations, such as multi-vehicle collisions. Moreover, although the paper focuses on simulation and test-track environments, a potential future step would be to extend these 118 test cases to more complex virtual or real-world settings. In doing so, a robust library of field-validated scenarios could serve as a shared reference for both industry and regulators.

A final challenge is the scalability of kernel-based methods: these can become computationally expensive for very large N . Although the paper presents approximations and theoretical insights, it generally presupposes that certain efficiency or approximation strategies can be used for big data.

Additional minor comments:

Explain the use of Accident-Rate-SUT as a performance metric.

The Discussion section (page 8) is somewhat weak; stronger elaboration on the results would improve clarity.

Algorithm 1 (page 11) needs more explanation.

In the theoretical discussion (page 12), consider providing a brief proof to enhance transparency.

(Remarks on code availability)

Reviewer #3

(Remarks to the Author)

In this paper, the authors present an algorithm for sampling test cases from a large test pool for the testing of autonomous driving systems. Experimental results demonstrate that the test cases sampled by the algorithm have high representativeness as well as coverage.

One major issue of the paper is the logic of presentation. The methodology is presented as the last section of the paper (page 9 to 13), whereas results are given right after Introduction (page 3 to 8). Discussion is provided in between (page 8 to 9). No conclusion is provided. Such a structure of the manuscript makes it very difficult for readers to understand the work.

In terms of technical details, it is surprising to see "single-vehicle conflicts are excluded" as the authors conjecture that "they primarily result from human errors." As a matter of fact, the current automated driving models are not immune from single-vehicle conflicts; for example, there are reports on single-vehicle conflicts caused by Tesla's Autopilot. The authors must provide strong rationale behind this setting (excluding single-vehicle conflicts).

It may be an overclaim to say "our approach effectively achieves both representativeness and coverage objectives" based on the results given in Table 1 (page 6). The approach gives a good trade-off between representativeness and coverage, instead of achieving both of them. From Table 1, we can observe that some baseline technique(s) outperformed the proposed approach in some metrics, while other baselines outperformed in other metrics. The proposed approach just gave a good balance among these metrics.

For the results of statistical realism, can quantitative measurement be given? Currently, the realism is visualized (through Figure 5 and 6) in a qualitatively way. It would also be worthwhile to compare the presented approach's realism with those of baseline techniques, like what have been done for the results in "Performance Comparison" (Figure 3 and 4, and Table 1).

(Remarks on code availability)

The code is just to generate some figures/table. It is not so relevant to the technical details of the proposed algorithm.

Reviewer #4

(Remarks to the Author)

- Regarding the safety argumentation

Other existing or under development safety argumentation frameworks deep dive into the approach itself, where scenario sampling is just one step. For instance, the Safety Assurance Framework (SAF) of the European research project SUNRISE, proposes a variety of steps that go far beyond scenario sampling. These include the selection of abstract and logical scenarios, the concretization of scenarios, the allocation of scenarios into test instances, the preparation of test cases (including the scenario + the function under test), the execution of tests (in virtual and real conditions), the evaluation of the results, and the potential consideration of in-service monitoring. In comparison, this paper's argumentation, whose main focus is the optimization of test case sampling, lacks a broader lifecycle management of the safety of ADAS/AD functions. As a consequence, in my opinion, it would be difficult to consider the methodology proposed in this paper as a complete and comprehensive solution for managing the safety lifecycle of ADAS/AD functions, especially in complex contexts such as the European automotive ecosystem, also considering the geographical and temporal limitations of the underlying SHRP2 NDS dataset. That said, the method proposed fits very well with certain steps of the SUNRISE SAF, providing a sound and efficient mechanism to select (sample) scenarios to be executed in the test instances, which could be a valuable tool within the scenario selection or test case preparation phase of the SAF

- Regarding the validity of the SHRP2-NDS

How might the dates when the SHRP2 NDS data was collected (2010-2013) impact its current relevance (2025) for ADS validation, given advancements in vehicle technology and availability of ADAS in modern cars?

To what extent do the driving behaviors and traffic scenarios within SHRP2 NDS still accurately reflect present-day real-world driving conditions?

Given that SHRP2 NDS represents US driving characteristics, are there limitations to applying scenarios from this dataset to

evaluate ADS in European or other non-US driving environments?

Are these SHRP2 NDS aspects anyhow affecting the proposed KTCS approach? (or it is completely unrelated?)

- Overall comments

The paper is of high quality in terms of formatting and language, showcasing careful proofreading and editing. All the references and figures within the document are consistent and logically integrated throughout the text, which greatly help the reader to understand the message.

Figures are detailed and built as rich infographics that help the reader to understand concepts easily.

(Remarks on code availability)

The code is provided as an online script with all the data necessary to reproduce the results and figures.

It includes Python routines to load the data, extract the statistics, and plot the results.

It also contains a version history of previous results.

Overall, the code presented supports well the argumentations on the paper.

Version 1:

Reviewer comments:

Reviewer #1

(Remarks to the Author)

Thank you for addressing my previous points. My only remaining major point concerns the optimised function $w_{\theta_{opt}}$, where θ is a p -dimensional vector. Both the function and θ are discussed in the Methods section, but are not mentioned again in either the Results section or the Supplementary Material. What is the value of p in practice? (I suppose $p \ll N$). Which θ vector has been estimated from the data?

Does the estimated θ vector provide any information, or do we simply have to trust it? It is also unclear whether p can be any number or if it must be selected within a given range.

Minor but important points:

Page 3: SUT is used without being defined. I also believe SUT and ADS are used to refer to the same thing. At least in page 3.

Methods section

- d must be defined soon after R^d and not in the following paragraph.

- The sentence "Ideally, this subset z_1, \dots, z_M contains both common cases and corner cases, as entire sets, the probability distribution function, constructed based on the selected subsets z_1, \dots, z_M , approximates the distribution of x_1, \dots, x_N " is not clear and has some problems with the English. Please consider to use two sentences. I would also remove "Ideally" because that must happen in practice and not (only) ideally.

Evaluation metrics section

- The sentence "Consider in space X " is not clear or wrong. Maybe the "in" should be removed?

- What kind of space is X ? Please clearly define things the first time they are introduced.

- A ")" is missing in the exponential formula (also elsewhere)

- σ in the exponential formula is not defined.

- x_i initially belong to R^d but later they belong to X . Does it mean that R^d and X are the same? This is very confusing for the reader.

- The z_i are sometimes bold and some times not.

- v_z^2 becomes v_z in equation (2) and elsewhere. Is this wanted?

- "Here, we use the term P_x ". Remove term because P_x is not the term of an equation but you are defining it.

- "coverage of the data space" (page 4) and "coverage of the data sample" (page 5) are not the same things but they are used like they are. I would use "data space" everywhere.

- MMD is called criterion and metric in two nearby sentences. Please only use metric to avoid confusion with common statistical criteria.

Stochastic Sampling Algorithm section

- The sentence "One way to generate stochastic samplings from the candidate test case pool" gives the idea that the x_i are candidate and can change. But I believe the x_i are fixed.
- In equation (5) S must be s .

KTCS Algorithm section

- Where are the weights w in step 1 of (11)? Is it correct that they are not used in the double sum?

Theoretical Discussion

- "the KTCS method can achieve sufficient representativeness because the constructed density distribution using the selected cases can approximate the original test case pool". The use of "can" gives the idea that those things may or may not happen.

(Remarks on code availability)

Reviewer #3

(Remarks to the Author)

Most of my comments in the previous round of review have been well addressed. However, I still have the concern about the exclusion of "single-vehicle conflicts". It seems that the authors agree with my argument and thus changed the rationale why single-vehicle conflicts were excluded. This is acceptable, but it brings another issue: The authors should explicitly claim the limitation of the scope of the paper, that is, the proposed method is limited to multi-vehicle conflicts only. This limitation should be reflected up front from the paper title.

(Remarks on code availability)

Although some extra data (example input and expected output) were added, there is still no code showing the implementation of the proposed algorithm.

Reviewer #4

(Remarks to the Author)

The authors have addressed all my previous comments and suggestions and added valuable remarks to the manuscript. The authors have debated on my questions on validity of the dataset, applicability in non-US environments, and compatibility with other safety argumentation initiatives. As a result I am convinced the readers will appreciate the paper and learn from it.

(Remarks on code availability)

I reviewed the code in the previous review session and I found it complete, clear (with traceability), and supporting the claims. Helpful for further research.

Version 2:

Reviewer comments:

Reviewer #1

(Remarks to the Author)

Thanks for addressing the additional points.

(Remarks on code availability)

Reviewer #3

(Remarks to the Author)

My concerns raised in the last round of review have been addressed in this revised version. I have no more comment.

(Remarks on code availability)

Current version of the code is fine.

Response to comments

“Optimizing Test Case Sampling for Safety Validation of Automated Driving System with Naturalistic Driving Study” (NCOMMS-25-16414)

We express our sincere gratitude for reviewers’ valuable and constructive comments. We have carefully revised the paper based on these comments and presented a point-by-point response addressing each comment. The comments are presented in italics, followed by our response. Please note that all page numbers referenced in this response correspond to the revised version of the manuscript.

Response to the reviewer 1’s comments

The article proposes a novel approach for selecting test cases from naturalistic driving studies for safety validation of automated driving system. The approach relies on the kernel embedding (of distributions) method and is based on an algorithm for test case selection that tries to achieve both coverage and representativeness of the multivariate distribution density of the driving features. The approach is then applied to 118 cases sampled from the Second Strategic Highway Research Program.

Comment 1. *In statistics, any inferential problem aims to analyse a sample taken from a population in order to answer relevant questions about the population itself. In this article, much effort is put into obtaining a sample that can represent another sample (which is itself a sample of a larger sample). Specifically, the 55,920 driving segments are a sample of the original 300,000 driving segments, which are a sample of SHRP2, which is a sample of the population of all cars in circulation in the US. Proving that the final 118 selected cases are a good approximation of the 55,920 driving samples does not prove that these cases are representative of the population. At the very least, it would be important to quantify the uncertainty of all results, bearing in mind that these 300,000 segments are neither the full SHRP2 dataset nor the population.*

Response: Thank you for the comment. We appreciate the opportunity to clarify the

design of the data collection process and the sampling procedure of the driving segments. The candidate driving case pool is drawn from the extensive and well-validated SHRP2 NDS driving dataset. Given the large volume of available data, a systematic sampling mechanism was employed to ensure that the selected segments closely approximate population-level driving characteristics. Our work is methodologically focused and aims to strike a balance between representativeness and computational feasibility. The proposed method can be readily extended with additional data to further broaden the reference population. Detailed responses are provided below.

First, the SHRP2 NDS is a large-scale data collection study (Dingus et al., 2016), systematically designed to account for key risk factors and potential confounders, such as driver demographics (e.g. age), vehicle type and geographical diversity (Hankey et al., 2016). Further details on how SHRP2 NDS achieves national representativeness can be found in Dingus et al. (2015). Given that the full SHRP2 NDS contains more than one million hours of continuous driving data, it is impractical to analyze the entire dataset directly. Therefore, a systematic design is implemented to sample test cases for analysis in a way that preserves the driving characteristics.

Secondly, we adopt a systematic sampling mechanism to construct the candidate pool of test cases, resulting in a good approximation of population information (Guo, 2019). The test cases within the candidate test case pool can be categorized into two groups: normal safe driving events and safety-critical crash events.

- (1) Normal driving segments are selected using a **two-stage stratified sampling** method that preserves the original driver population characteristics and mitigates selection bias across drivers. In the first stage, the number of segments drawn from each driver is determined proportionally to their total driving time. In the second stage, a fixed number of segments is randomly sampled from each driver’s data, resulting in approximately 300,000 segments and covering around 50 million meters of driving.
- (2) Safety-critical events are selected from the full population of crash data in the SHRP2 NDS. Using a validated crash identification method (Hankey et al., 2016), all

crash events are identified and incorporated into the candidate test-case pool.

We further clarified the sampling strategy on page 8 of the revised manuscript, as highlighted in red below.

A systematic sampling scheme is used to construct the test case pool based on traffic crash events and normal driving segments extracted from SHRP2 NDS, ensuring a close approximation of population-level driving characteristics. The test cases are categorized into two groups: normal driving and safety-critical events. Each case consists of a 15-second driving segment, treated as a driving scenario. The selection of normal driving cases follows a two-stage stratified design that preserves the original representation of the driver population and mitigates selection bias across drivers (Guo, 2019). First, the number of normal driving segments is determined proportionally by the total driving time of each driver. Then, a preset number of driving segments is randomly sampled from each driver’s data. In total, about 300,000 randomly selected 15-second driving segments were chosen, covering approximately 50 million meters of driving from the SHRP2 NDS database. Safety-critical events are selected from the full population of crash data in the SHRP2 NDS. Using a validated crash identification method (Hankey et al., 2016), all crash events are identified and incorporated into the candidate test-case pool.

Our study is primarily methodological, using the SHRP2 NDS to demonstrate the effectiveness of the proposed framework with a hypothetical baseline reference population. This baseline serves as a starting point; over time, the reference population can be expanded by incorporating additional industry data obtained through large-scale crowdsourcing initiatives. Since drawing inferential conclusions about the broader driving population from a multi-stage sampling design requires careful treatment of uncertainty, we quantify this uncertainty by computing confidence intervals for the resulting safety metrics. Further discussions are provided in our response to Reviewer 2, Comment 2 and Reviewer 1, Comment 5.

Comment 2. Equation (5) introduces the variance in a one dimensional space and its role to achieve coverage is subsequently discussed. Then the same concept is extended to the multivariate case. When dealing with multivariate distributions, however, covariances become important. These are completely ignored in the definition of information potential in equations (6) and (7). Ignoring covariances likely implies that the coverage is reached in a sub-optimal way.

Response: Thank you for the comment. As noted in the manuscript, Equation 5 (Equation 1 in the revised manuscript) introduces variance in a one-dimensional space, and its role in achieving coverage is discussed. We then extend the same concept to the multivariate case, where covariances between features indeed become important. We address this by adopting a nonparametric approach within the Reproducing Kernel Hilbert Space (RKHS) framework. Specifically, the multivariate variance is defined in the RKHS, rather than in the original feature space. The RKHS offers desirable properties to capture nonlinear relationships and higher-order interactions between features (Muandet et al., 2017). It has been widely applied to complex feature covariances in data types such as 2D images, 3D tensor data, and graph-structured data (Li et al., 2017; Chen et al., 2019; Han et al., 2024).

The test case pool in our analysis considers the interactions among the driving kinematic features. Following the approach outlined in Wolfer and Alquier (2025), we define the multivariate variance in RKHS as

$$\nu_z^2 = \mathbb{E}_{z \sim \mathbb{P}} \|\mathcal{K}(z, \cdot) - \mu_{\mathbb{P}}\|_{\mathcal{H}}^2 \quad (\text{R.1})$$

This formulation captures the dispersion of the multivariate variable $\mathbf{z} \in \mathbb{R}^d$ in RKHS. Here, $\mu_{\mathbb{P}} = \int \phi(\mathbf{x}) d\mathbb{P}(\mathbf{x}) = \mathcal{K}(\mathbf{x}, \cdot)$ represents the kernel mean embedding, which encodes both the marginal distributions and the covariance structure among the features. This is achieved by using a kernel function to map the high-dimensional data \mathbf{x}_i into RKHS (Muandet et al., 2017).

This approach allows us to avoid the need to directly estimate the covariance matrix of the input features, which can be unstable and unreliable in high-dimensional settings

(Pourahmadi, 2013). In particular, both the sample covariance matrix and its inverse often suffer from instability when dealing with large numbers of features. Instead, we handle feature covariance information through the use of kernel functions.

The kernel function $\mathcal{K}(\mathbf{x}_t, \mathbf{x}_s)$ measures the similarity between cases \mathbf{x}_t and \mathbf{x}_s by evaluating all feature coordinates simultaneously. If \mathbf{x} and \mathbf{x}_s differ significantly along any correlated direction, the kernel value $\mathcal{K}(\mathbf{x}_t, \mathbf{x}_s)$ will be small. This method not only captures individual feature differences but also considers how features vary together, thus encoding their covariance structure (Simon-Gabriel and Schölkopf, 2018). By aggregating these kernel similarities, the Information Potential (IP) approach implicitly incorporates means, variances, and covariances. This ensures that coverage is assessed on the joint distribution of investigated features, rather than in a purely marginal sense (Principe, 2010).

We revised the manuscript on page 4 (in the revised manuscript) to explain how the RKHS method addresses the high dimensionality of the feature space and nonlinear dependencies among the features.

Considering the potentially high dimensionality of the feature space (d) and the nonlinear dependencies among the features, we conduct our analysis in the reproducing kernel Hilbert space (RKHS) Gretton et al. (2012). As a nonparametric framework, RKHS avoids the need to explicitly estimate the high-dimensional covariance matrix of the input features, a procedure that is often unstable and unreliable in such settings Pourahmadi (2013). Beyond this practical advantage, RKHS is well suited to capture nonlinear relationships and higher-order feature interactions, and has been successfully applied to data with complex covariance structures Li et al. (2017); Han et al. (2024).

The kernel function $\mathcal{K}(\mathbf{z}, \mathbf{z}')$ measures the distance between cases \mathbf{z} and \mathbf{z}' by evaluating all the feature coordinates simultaneously. This allows pairwise similarities to embed second-order dependencies among the features in the RKHS (Simon-Gabriel and Schölkopf, 2018). By aggregating these kernel similarities, the Information Potential (IP) approach implicitly incorporates moments of all

orders, such as means, variances, and covariances. This ensures that coverage is assessed on the entire joint distribution of the features, rather than just the marginal distributions (Principe, 2010).

In addition, we conducted numerical simulations in the Supplementary Material (pages 1-4) to validate that our proposed method can effectively handle correlated feature settings. Three types of simulation examples are considered that cover various covariance structures of characteristics, different sample sizes, and different feature dimensions. The numerical simulation results consistently show that, across various correlated high-dimensional simulation setups, our method performs well in terms of coverage ability.

We also discuss the optimization in terms of coverage. Theorem 1 and Theorem 2 establish the convergence properties of the KTCS algorithm with respect to its coverage ability. As the candidate pool grows sufficiently large, the IP of the subset selected by KTCS converges to the global minimum of the IP objective, thereby achieving the goal outlined in Step 1 of the KTCS algorithm.

We hope this addresses your concern. Thank you again for your constructive feedback and thoughtful comments.

Comment 3. *Page 11: The definition of the weights w_i used in equation 11 is rather obscure and needs to be clarified. As I understand it, the weights are a function of the observed features used to minimize the information potential. What is the implication of this choice?*

Response: Thank you for the comment. This choice not only allows us to assign an individual weight w_i to each case \mathbf{x}_i , but also reduces computational expenses by using a shared parametric function through $w_i = w_\theta(\mathbf{x}_i)$.

First, weights w_i are assigned to each of the case \mathbf{x}_i so that Pareto-order sampling (Tillé, 2020) can be implemented to select exactly M test cases from $\mathbf{x}_1, \dots, \mathbf{x}_N$. Since selecting a subset of size M from a large candidate set of size N is an NP-hard problem, Equation 11 (Equation 7 in the revised manuscript) reformulates the problem by introducing continuous, non-negative weights w_1, \dots, w_N , which are learned to minimize the information potential.

Each candidate test case \mathbf{x}_i is assigned a weight w_i , resulting in $\{(\mathbf{x}_1, w_1), \dots, (\mathbf{x}_N, w_N)\}$. Larger values for w_i indicate higher chances of being selected among the M selected cases.

Second, to improve computational efficiency, we set $w_i = w_\theta(\mathbf{x}_i)$ through a shared parametric function. To keep the optimization in Equation 10 (Equation 6 in the revised manuscript) tractable, we avoid treating w_1, \dots, w_N as independent free parameters to be optimized. Instead, each weight w_i is generated by the shared parametric function $w_i = w_\theta(\mathbf{x}_i)$, meaning the weight of a point depends only on its own observed feature vector. Since θ has fewer degrees of freedom than N , this approach makes optimization tractable while still allowing the weights to adapt to the feature distribution.

We added the following discussion in the revised version of the paper. Revisions are colored in red.

We set the importance measure for case \mathbf{x}_i as w_i , which results in a weighted set $\{(\mathbf{x}_1, w_1), \dots, (\mathbf{x}_N, w_N)\}$. Larger values in w_i indicate a higher chance of the corresponding case \mathbf{x}_i being among the M selected cases. Rather than treating w_1, \dots, w_N as independent free parameters to be optimized, which would be computationally expensive, we approximate their optimal values using a shared parametric function. That is, we set the importance measure for the case \mathbf{x}_i as $w_\theta(\mathbf{x}_i)$, where the function is parameterized by θ as $w_\theta : \mathbb{R}^d \rightarrow \mathbb{R}$.

Comment 4. *The KTCS algorithm is based on two steps. The first step tries to achieve coverage (maximizing the variance) while the second step tries to achieve representativeness by minimizing the distance between the distribution of the smaller sample with the distribution of the larger sample. The two steps are not independent because a distribution is “defined” by its moments, including its variance. What I see in this case, is a first step where M cases are selected (in a sub-optimal way because covariances are ignored) and then some weights (the lambdas) which are supposed to “reflecting the importance of each scenario in driving” (end of page 2). What I see is that the lambdas simply try to adjust what is done in step 1, where the actual “shape” of the distribution density in the high dimensional space is ignored because the variance on its own cannot describe it. Also, I do not see how these*

weights are related to the importance of the driving scenarios. The same weights are used in Equation (3) to define the accident rate so they have an important role, but I do not see a validation of this weights and their meaning (also because they have been computed on only 118 cases). I also suspect that those weights are “responsible” for the problem discussed in my next point.

Response: Thank you for the comment. (1) The optimization is performed using non-parametric kernel methods, computed over the joint distribution of the driving data, including both variance and covariance. Further details refer to the Reviewer 1’s Comment 2 and the revised manuscript on Page 4. (2) We rephrased the usage of the λ -weights as distribution alignment, rather than indicators of “scenario importance”. (3) We provided numerical experiments through ablation studies to validate that the λ -weights accurately reflect the actual shape of the distributions and address the concerns of “simply try to adjust what is done in step 1”.

First, we review the overall steps of the KTCS algorithm, highlighting that Step 1 and Step 2 are complementary. Since the optimization aims to simultaneously optimize representativeness and coverage, which constitutes a multi-objective optimization problem, we choose to optimize coverage first and then adjust it to optimize the representativeness. Specifically, the optimization objectives differ between the steps. Step 1 focuses on a discrete design, where the key idea is to select M subcases from the original N -sized candidate test case pool. The optimization variables are w_1, \dots, w_N for $\mathbf{x}_1, \dots, \mathbf{x}_N$, and the objective is to minimize the IP, which quantifies coverage. In Step 2, we re-weight the M selected cases so that their empirical distribution aligns as closely as possible with that of the full candidate pool. The optimization variables are $\lambda_1, \dots, \lambda_M$ for $\mathbf{z}_1, \dots, \mathbf{z}_M$, and the objective is to minimize the MMD, which quantifies representativeness.

Secondly, regarding the meanings of $\lambda_1, \dots, \lambda_M$, these weights are used to realign the empirical distribution of the subset $\mathbf{z}_1, \dots, \mathbf{z}_M$ with that of the entire candidate test case pool $\mathbf{x}_1, \dots, \mathbf{x}_N$, complementing the coverage achieved in Step 1 and ensuring complete representativeness. Gretton et al. (2012) shows that when the value of MMD is sufficiently small, it implies that all corresponding moments (means, variances, and higher-order moment com-

binations encoded by the kernel) are identical. Given this, minimizing the MMD helps to force the weighted version of the selected cases to reflect the actual shape of the distributions. Importantly, these λ -weights are not derived from the 118 cases in isolation; they are computed from the pairwise kernel similarities between the 118 scenarios and each of the 55,920 candidates, and are chosen precisely to minimize MMD.

To avoid confusion with the importance weights w_i used in Step 1, we rephrase the role of the λ -weights in Step 2 as follows in the Introduction section.

To achieve this, an attention mechanism is introduced that assigns distribution-alignment weights to the selected cases (Vaswani et al., 2017; Chai and Wang, 2022), adjusting the selected subset so that its empirical distribution matches that of the full candidate test case pool.

We added the following discussion in the Methods section.

Thus, we can obtain the selected points $\mathbf{z}_1, \dots, \mathbf{z}_M$ and the associated weights as $\lambda_1, \dots, \lambda_M$. These λ -weights serve as distribution-alignment weights, used to realign the empirical distribution of the subset $\mathbf{z}_1, \dots, \mathbf{z}_M$ with that of the entire candidate test case pool $\mathbf{x}_1, \dots, \mathbf{x}_N$.

To validate $\lambda_1, \dots, \lambda_M$, we conduct the following ablation studies, particularly highlighting their ability to approximate different moments of the distributions. To address your concerns about whether “ λ s” simply adjust what is done in Step 1, we replace the “ λ -weights” with the w -weights obtained in Step 1. The results are reported in the column “ w -weights Alignment” of Table R.1. We added the corresponding content in the Supplementary Material. The related content is directly quoted as follows.

We conduct the following ablation studies to validate the role of the λ -weights in the proposed KTCS algorithm. The weights $\lambda_1, \dots, \lambda_M$ realign the distributions of the M selected cases with that of the original N -sized candidate test case pool. To quantitatively assess the contribution of Step 2 to the overall approximation

quality, we consider two additional variants of the KTCS algorithm: (1) No Alignment: No distributional alignment is applied; each case is equally weighted. (2) w -weights Alignment: The λ -weights are replaced with the w -weights derived in Step 1 of the KTCS algorithm. The performances of the MMD, ADD and H-dist metrics are reported in Table R.1.

To assess whether our method can capture the true shape of high-dimensional distributions, particularly different moment information, we measure discrepancies at three moment levels for each setting: (1) Mean Discrepancy (Mean Dis): Measures the difference in mean values, calculated as $\|\mu_M - \mu_N\|_2$, where μ_M and μ_N are the mean vectors of the M selected cases and the N -sized candidate pool, respectively; lower values indicate closer means. (2) Variance Discrepancy (Var Dis): Measures the difference in variance across dimensions, calculated as $\frac{1}{d} \sum_{i=1}^d |\sigma_{M,i}^2 - \sigma_{N,i}^2|$, where $\sigma_{M,i}^2$ and $\sigma_{N,i}^2$ are variances along the i -th dimension; lower values indicate more similar variances. (3) Covariance Discrepancy (Cov Dis): Measures the difference in covariance structures, calculated as $\|\Sigma_M - \Sigma_N\|_F / \|\Sigma_N\|_F$, where Σ_M and Σ_N are the covariance matrices constructed from the M selected cases and the candidate pool, respectively, and $\|\cdot\|_F$ denotes the Frobenius norm; lower values indicate more similar covariance structures. The reports are presented in Table R.1.

Table R.1: Ablation study of the KTCS algorithm.

Metric	λ -weights Alignment	No Alignment	w -weights Alignment
MMD ↓	0.020	0.286	0.382
ADD ↓	0.018	0.317	0.442
H-Dist ↓	1.007	3.877	5.079
Mean dis ↓	0.091	0.644	1.559
Var Dis ↓	0.003	0.030	0.042
Cov Dis ↓	0.035	0.255	0.523

Note: ↓ indicates that lower values are better. H-dist is an empirical distance measure that can take negative values (Zhang et al., 2023).

Table R.1 shows that Step 2 of the KTCS algorithm decreases distributional distances (MMD, ADD, H-Dist) by an order of magnitude and substantially

improves moment-matching accuracy.

Lastly, we respectfully clarify that the λ -weights appearing in Equation 3 (Equation 17 in the revised manuscript) are exactly those derived in Step 2 of the KTCS algorithm. As shown in Table R.1, the λ -weights obtained from Step 2 perform moment matching in an RKHS constructed from the joint feature distribution. The optimization inherently accounts for both variances and covariances, thereby preserving the true shape of the distribution from the candidate test case pool. Consequently, the final accident-rate estimator remains unbiased with respect to the full candidate pool (see derivation in our response to Reviewer 1, Comment 5).

Comment 5. *When it comes to apply the methodology to quantify the safety level of an ADS the results do not seem to be satisfactory. The scaling risk in Figure 7 goes from 0.0001 to 13.8819 depending on which the 5 failed cases are. How is your approach useful in practice if the scaling risk may change by a factor of 138,800 by changing the trial? I believe the case study is too weak, and/or it shows that 118 cases are not enough for the safety validation of the driving system (which is the claimed goal of the approach).*

Response: Thank you for the comment. The scaling risk (SR) in our study is defined as the ratio of the System Under Test’s (SUTs) crash rate to the crash rate of benchmark human drivers. In other words, it is a relative risk measure that compares the safety performance of the SUT with that of human drivers. The benchmark human crash rate we adopt, estimated from the SHRP2 NDS, is extremely low, approximately 1.7×10^{-9} . When such a small number appears in the denominator, even a slight absolute change in the SUT crash rate can lead to a large change in the resulting scaling risk. This is a mathematical characteristic of using a ratio when the denominator is near zero, a phenomenon commonly observed in transportation safety research (Guo, 2019). Therefore, the large variation in SR across trials is not indicative of an issue with the methodology itself but is rather a result of the inherently low benchmark crash rate.

We added the following content in the revised manuscript to further discuss SR and its .

The scaling risk (SR) in our study is defined as the ratio of the SUT’s crash rate

to the crash rate of benchmark human drivers, particularly in terms of being involved in severe traffic crashes. In other words, it is a relative risk measure that compares the SUT’s safety performance with that of human drivers. The SR can be calculated as the follows:

The wide spread in SR values shows that the SUT’s measured safety highly depends on which test cases the SUT fails. Each of the 118 selected scenarios carries a λ -weight. Each selected test case can be regarded as a representative case from similar ones, meaning that evaluating the selected cases provides insights into the system’s performance in similar situations with the minimized MMD. A larger weight λ_i for case z_i means that many similar situations occur in the pool, so failure in that case z_i has a larger influence on the overall metric. The λ -weights themselves span widely, from 6.983×10^{-8} to 0.165 (see Figure 1 and 2 in the Supplementary Material). This wide range directly drives the corresponding variability in SR estimates. We add the following discussion to explain the interpretations of the scaling risk. The revisions are highlighted in color red.

In contrast, in Figure 8(a), the five failed cases (ID: 63, 80, 91, 32, 35) result in safer performance compared to human drivers. All five of these cases are associated with weights smaller than 0.001. Notably, these cases share common characteristics: large values in *accz-std*, *xpos-min*, and *ypos-min*, suggesting a bumpy driving scenario with few nearby vehicles. Such conditions are rare in real world settings, and their low weight reflects their limited impact in determining the overall safety risk of the SUT. The wide range of scaling risk values (0.0001 to 13.38 in Figure 8) does not mean the ADS’s risk literally fluctuates within this range during a single evaluation. Instead, it indicates that depending on which specific scenarios the ADS fails, the interpretation of the test results could be different. In this demonstration, the extreme values (0.0001 and 13.38) reflect the best and worst case combinations among the five selected failure cases.

Next, to better quantify the uncertainty of SR which is a point estimator, we derive its corresponding confidence interval. This confidence interval quantifies how much information

the data contain about the true risk and supports formal statistical hypothesis testing. We add the following discussion for the SR. The detailed proof is attached in the supplementary material.

We additionally derive SR’s corresponding confidence interval to better quantify the uncertainty of SR. Specifically, the 95% confidence interval for the log value of the scaling risk can be constructed as the follows:

$$\log SR \pm 1.96 \times \sqrt{\frac{\sum_{j=1}^M \lambda_j^2 r_j (1 - r_j)}{(\sum_{j=1}^M \lambda_j r_j)^2}} \quad (\text{R.2})$$

where $r_j = (I_j + 0.5)/2$. If the lower bound of the established confidence interval larger than 1, then we have 95% confidence to say the SUT could be riskier than human drivers.

To validate the consistency of the proposed measure, we treat SHRP2 drivers as the SUT by letting I_j represent crashes in each test case. Given the SHRP2 NDS baseline accident rate, the resulting ratio should be close to 1 and the value of 1 should be within the 95% confidence interval if our calculations are accurate. Using data from the 118 test cases, along with their associated weights and distances, the calculated accident rate for the “SHRP2-as-SUT” scenario is 1.993×10^{-9} crashes/meters. This results in a SR value of 1.176, the 95% confidence interval [0.441, 3.138], the 95% confidence interval [0.441, 3.138], proving that our method is statistically validated. For Figure 8(b), out of the 118 selected cases, the point estimation of the scaling risk is 13.8819, the 95% confidence interval of [6.53, 29.50], indicating that, under this failure pattern, the SUT is significantly riskier than human drivers.

We fully acknowledge that 118 test cases is a limited sample, and ideally more testing would reduce the uncertainty in the safety assessment. Unfortunately, each test is extremely resource-intensive: a single high-fidelity simulation or on-road scenario can cost thousands of dollars (Scanlon et al., 2021), and recreating the same scenario in a controlled track

environment is even more expensive (Albrecht et al., 2021). Consequently, even well-funded programs rarely exceed a few hundred runs (Feng et al., 2023). Our choice of 118 cases was thus bounded by what is feasible in practice with available time and budget. The reason for this specific number is explained in response to Reviewer 1, Comment 22.

In summary, the large variability in scaling risk stems from the combination of an extremely low baseline crash rate and the weighting of diverse scenarios in a limited test set. This variability is informative rather than merely problematic: it tells us that the outcome (safety level) highly depends on which specific scenarios the ADS struggles with. Our approach remains useful in practice because it provides a structured, data-driven way to quantify safety and highlight critical areas for improvement. Our analytical framework is expressly designed to be scalable, new scenarios or larger sets of test cases can be added at any time, and the SR/CI recomputed, without changing the mathematics.

Comment 6. *Page 1: “As shown in Figure 1(a), these assumptions can diverge significantly from actual on-road conditions”. Figure 1(a) only shows some invented examples which (I guess) are not taken from the SHRP2 dataset.*

Response: Figure 1(a) uses illustrative (hypothetical) scenarios to highlight the motivation and challenges associated with our research question. To address your concern, we added an actual test case from the SHRP2 NDS to the supplementary material and discuss it as follows:

Figure R.1 shows an example raw test case from the SHRP2 NDS dataset. Panels R.1 (a)-(c) present the subject vehicle’s three-axis accelerations, longitudinal acceleration (acc_x), lateral acceleration (acc_y), and vertical acceleration (acc_z), as driving dynamics over a 15-second interval. Panel (d) then shows the resulting speed profile, increasing from approximately 1 m/s to about 18 m/s, which corresponds to the positive acceleration observed in panels (a)-(c). Panels R.1 (e)-(g) characterize the surrounding traffic detected by the radar sensors: (e) shows the trajectories of six surrounding vehicles over the same 15-second window, with each color representing a different vehicle, while (f) and (g) display the relative longitudinal and lateral speeds, respectively, between the subject vehicle

Figure R.1: **Example of the raw data for one test case in our candidate test case pool.** (a–c) Time-series of the subject vehicle’s longitudinal ($accx$), lateral ($accy$), and vertical ($accz$) accelerations. (d) Driving speed of the subject vehicle. (e) Trajectories of surrounding vehicles. (f) Relative longitudinal speed (g) relative lateral speed.

and each of the six surrounding vehicles.

Comment 7. *Page 1: The Second Strategic Highway Research Program is introduced without a reference.*

Response: Thank you for the comment. We add reference on Page 2 (in the revised manuscript) as the follows:

To construct the test case pool for safety validation, we adopt the largest-scale naturalistic driving study (NDS), the Second Strategic Highway Research Program (SHRP2) (Dingus et al., 2016)

Comment 8. *Page 2: I believe “Too much focus on common cases” and “Too much focus on corner cases” are swapped in Figure 1(c).*

Response: Thank you for the comment. We revise it correspondingly in Figure R.2.

Comment 9. *Page 2: “Representativeness ensures that the cases collectively approximate the distribution of real-world scenarios”. At this point in the article, the distribution (of driving characteristics) has not yet been defined, so it is not clear which distribution it*

Figure R.2: **Schematic diagram of test case selection for safety validation of automated driving systems.** (a) Overview of the test case selections for ADS safety validation. (b) The study utilizes naturalistic driving study data from SHRP2, which provides high-quality, comprehensive driving information. SHRP2 NDS collected data from six U.S sites, providing a unbiased benchmark for human driver driving data. (c) Selected test cases guarantee both representativeness and coverage for effective validation.

is.

Response: Thank you for this helpful comment. To clarify, we revised the sentence to explicitly explain what we mean. The revised version is as the follows:

Representativeness ensures that the selected cases, taken together, **reflect the realism of actual driving situations observed on public roads (as derived from the SHRP2 NDS).** Using these cases, we can create a test set that **realistically mirrors how often different driving scenarios occur, enabling more meaningful and credible safety evaluations.**

Comment 10. Page 2: “a small number of strategically selected green points around the core of the testable space can effectively represent complex driving scenarios”. Complex in terms of what? I do not see why the cases around the core necessarily represent complex scenarios.

Response: To clarify, we revised the sentence in the manuscript as the follows. We

removed the term “complex” to avoid any confusion.

A small number of strategically selected green points around the core of the testable space can effectively represent **the majority of routine driving scenarios**.

Comment 11. *Page 2: “Coverage ensures that cases include diverse driving scenarios, particularly in low-density areas”. Similarly, density has not yet been formally introduced.*

Response: Thank you for this helpful comment. To address this, we revised the sentence to explicitly clarify what we mean by “low-density areas” as the follows.

Coverage ensures that selected cases include diverse driving scenarios, particularly those that are less frequent yet carry safety-critical information and might otherwise be missed, marked in red in Figure R.2(c). For example, sudden cut-in by a lead vehicle at high relative speed on a curved highway exit ramp is rare in naturalistic driving but highly critical for ADS safety.

Comment 12. *Page 2: “can effectively target high-risk areas but may lack fidelity”. What is fidelity? It is not formally defined in the article.*

Response: Thank you for the comment. To clarify this, we revise this as the follows:

can effectively target high-risk areas but may lack **representativeness**.

Comment 13. *Page 2: “To achieve this, an attention mechanism is introduced that assigns weights to the selected cases”. A reference about the attention mechanism is needed.*

Response: Thank you for the comment. We add the citations and the revisions are directly quoted as the following:

To achieve this, an attention mechanism is introduced that assigns weights to the selected cases (Vaswani et al., 2017; Chai and Wang, 2022)

Comment 14. *Page 2: “Higher weights indicate more common scenarios, while lower weights correspond to less frequent ones”. This interpretation of the weights is different from*

the interpretation elsewhere in the article, where a higher weight is associated with cases of higher importance. But being an important case is not related to being a frequent or less frequent case.

Response: Thank you for the comment. These weights $\lambda_1, \dots, \lambda_M$ are used to realign the empirical distribution of the subset with that of the entire candidate test case pool, as detailed in the response to Comment 4 of Reviewer 1. To clarify your concern, we revise the sentence as the follows:

A larger weight in the value assigned by the attention mechanism indicates that the corresponding selected case contributes more to approximating the underlying distribution of the candidate test-case pool.

Comment 15. *Page 3 first line: “performance based on the success or failure of each case”. What is a success and what is a failure?*

Response: Thank you for the comment. We add the following content to the revised version of the manuscript to clarify this concern.

performance based on the success or failure of each case, where a case is labeled a success if the SUT completes it without a traffic crash, and a failure if the ADS does not complete it safely and a crash occurs.

Comment 16. *Page 3 first line: “Scaling risk reflects how much more risk the ADS carries compared to human drivers”. Risk usually has a formal definition, depending on the context. What is risk in this case?*

Response: Thank you for the comment. We revise the sentence as the follows to clarify this:

Scaling risk reflects how much more risk the ADS carries compared to human drivers, specifically in terms of being involved in severe traffic crashes ¹

¹Severe traffic crashes here include police reportable and fatal crash according to definition in Hankey et al. (2016). Fatal crashes involve air-bag deployment, human injury, or a vehicle rollover. Police reportable crashes are those that incur at least \$1,500 in property damage.

Comment 17. *Page 3: “The proposed algorithm addresses the NP-hard problem of minimizing the information potential of driving data by assigning varying weights to selected test cases, thus ensuring comprehensive coverage of long-tailed critical driving scenarios.”*

What is a tail in a multidimensional space?

Response: Thank you for the comment. We revise the sentence as the follows:

his dual approach ensures comprehensive coverage of long-tailed, safety-critical driving scenarios—**which deviate substantially from the most common driving conditions**—while still matching naturalistic driving patterns.

For a detailed discussion of tail in the multidimensional space and how we identify this, please see our response to Reviewer 1’s Comment 28.

Comment 18. *Page 3: “To our knowledge, this is the first method to effectively use a small number of test cases to represent the entire real-world driving space while also providing theoretical guarantees for ADS safety validation.” I do not think this can be claimed, for the reason explained in my first major point.*

Response: Thank you for the comment. We revise the sentence as the follows:

To our knowledge, this is the first method to effectively use a small number of test cases **derived from nationwide large scale naturalistic driving study** to **strategically** represent **real world driving situations**, while also providing theoretical guarantees for ADS safety validation.

Comment 19. *Page 3: “We employed importance sampling to construct a test case pool from traffic crashes”. Written this way, it looks like the pool only contains crashes.*

Response: Thank you for the comment. We revise such sentences as the follows:

A systematic sampling scheme is used to construct the test case pool **based on traffic crash events and normal driving segments extracted from SHRP2 NDS, ensuring a close approximation of population-level driving characteristics.**

Comment 20. *Page 3: “The test cases are categorized into two groups: normal driving and safety-critical events.” What is normal driving? And what are safety-critical events? Just crashes or something else?*

Response: We revise the sentence as the follows to make it more precise:

The test cases are categorized into two groups: normal **safe** driving and safety-critical **traffic crash** events.

Comment 21. *Page 4: “In total, we randomly selected about 300,000”. What is the size of the full original pool?*

Response: In our analysis, each test case corresponds to a 15-second driving segment. Dividing the entire SHRP-2 NDS dataset (containing over 1 million hours of naturalistic driving data (Dingus et al., 2016)) into continuous 15-second segments would yield more than 240 million ($1,000,000h \times 3600s/h \div 15s$) test cases.

Comment 22. *Page 4: “We set $M = 0.5\sqrt{N}$, resulting in 118 selected cases for our analysis”. Why this choice of M ? And how do you know that the sample of M cases will include crashes? “In total, 55920 normal driving segments and 90 crash events are used for the analysis”, so the crashes are very small in number so I guess it is possible that the 118 cases do not include crashes?*

Response: We add the following content in the Supplementary Material to discuss the reason for the choice for number of selected cases.

In determining the number of selected cases, Theorem 3 demonstrates that choosing $M = \mathcal{O}(\sqrt{N})$ achieves desirable convergence properties: the M selected cases closely approximate the distribution of the full set of N candidates test cases. The setting $M = 0.5\sqrt{N}$ achieves satisfactory performance in both representativeness and coverage. Industrial-scale ADS safety evaluations typically conduct on the order of one hundred test runs due to real-world budget and time constraints. For instance, Feng et al. (2023) executed 117 test cases. Balancing this practical benchmark with our theoretical guidance, we choose $M = 0.5\sqrt{N}$, which is 118 selected test cases.

The KTCS algorithm selects M test cases from the candidate test case pool without using label information (i.e., whether a case is a crash or not). This label-agnostic approach ensures that the selection process is unbiased with respect to crash occurrence, relying instead on the distributional properties of the data. After the selection, we validate whether the selected cases belong to traffic crashes or not. The results show that 5 out of the 118 cases are identified as traffic crashes (case IDs are 7, 18, 45, 89, 96 in Figure 1 and 2 in the supplementary material). Based on these, we calculate the scaling risk and associated confidence interval. Ideally, the point estimator of SR should be close to 1, and the 1 should be within the constructed confidence interval, as the outcomes of these 118 cases with those observed for human drivers in the SHRP2 NDS. Following the aforementioned computation process, the SR is 1.176, the 95% confidence interval [0.441, 3.138], proving that our method is statistically validated and has sufficient coverage ability. Details are provided in the responses to Comment 5 of Reviewer 1.

Comment 23. *Page 4: “The density distributions are generated using kernel density estimation, with darker blue indicating higher density regions.” What about the white area? Figure 3 must include a color scale of the density.*

Response: Thank you for the comment. The white area indicates regions where the estimated density values are close to zero, based on the points selected by the corresponding method. We included an explicit color bar that quantifies the density scale of the density in Figure R.3.

Comment 24. *Figure 3: Why are the selected cases (red markers) only shown for the novel approach?*

Response: We updated in Figure R.3 to include red markers for all methods.

Comment 25. *Page 5: “MMD-Critic, on the other hand, focuses on a few long-tailed cases but lacks the ability to comprehensively represent safety-critical situations”. How can you tell this? What is a safety critical situation in terms of driving features?*

Response: Thank you for the comment. Safety-critical situations usually exhibit large variations in terms of driving dynamics. For example, the *accy-std* measures the variation

Figure R.3: **Comparison of density distribution based on points selected by different methods.** (a) Joint distribution of features *accy-std* and *xpos-min*. (b) Joint distribution of feature *speed-min* and *xpos-std*. (c) Joint distribution of feature *xvel-std* and *xpos-min*. The blue area represents the estimated density distribution, with darker shades indicating higher density. Red points mark the selected cases, and the size of each point reflects the weight of that case.

of lateral acceleration of the subject vehicle, as shown in Figure 4(b). These safety-critical situations typically lie at the edges of joint distributions, that is, in the long-tailed regions.

Figures 4(b) and 4(c) confirm that MMD-Critic can indeed select some samples far from the distribution center, thereby capturing part of the long tail. However, a substantial blue region remains uncovered by the red points (the cases selected by MMD-Critic). This uncovered region corresponds to long-tail driving scenarios that exist in the NDS but are not incorporated into the MMD-Critic test set, making it difficult to evaluate the ADS under those safety-critical conditions. Accordingly, we revised the sentence as follows:

MMD-Critic selects certain long-tail cases located far from the center of the investigated joint distributions, but it fails to provide comprehensive coverage of the entire long-tail region, as evidenced by the large blue area not spanned by the red points in Figure 4.

Comment 26. Page 5: “While MED demonstrates some capacity to cover long-tailed

cases, our method selects more evenly distributed points, particularly along boundary lines.”
What is a boundary line? In any case I do not see this as a good property. This is the result of maximizing the variances without considering the covariances.

Response: Thank you for your insightful comment. By *Boundary lines*, we refer to points located near the outer limits of the feasible scenario parameter space derived from the SHRP2 NDS. These points represent the most extreme or rare combinations of parameters that are most likely to reveal safety-critical behaviors. For example, the simultaneous occurrence of high speed and large driving variability creates challenging driving conditions that significantly increase the risk of vehicle instability, making such scenarios critical for evaluating safety performance.

Figure 4(c) shows that the MED-selected points cluster tightly in the tail, with many exhibiting nearly identical values of *ypos-std* (lateral-distance variability) and *speed-min*, thereby offering limited additional information. In contrast, our method distributes points more evenly throughout the tail and interior regions, reducing redundancy and enhancing test efficiency. We revised the Results Section as follows:

While MED demonstrates some capacity to cover long-tailed cases, our method selects points more evenly distributed, particularly near the outer limits of the feasible parameter space. As shown in Figure 4(c), MED-selected points cluster tightly in the tail, with many exhibiting nearly identical values of *ypos-std* (lateral-distance variability) and *speed-min*, thereby offering limited new information. In contrast, our method distributes points throughout both the tail and interior regions, reducing redundancy and enhancing test efficiency.

Regarding the feasibility of our optimization procedure, we discussed the treatment of covariance-related issues in our response to Comment 3 from Reviewer 1, and presented quantitative evidence supporting the accuracy of the covariance approximation in our response to Comment 4 from Reviewer 1.

Comment 27. *Page 6: “Corner cases, characterized by long-tailed characteristics”. Again, it is not clear what is a tail in a multidimensional space.*

Response: Thank you for the comment. Please refer to our response to Comment 17 and 28 of Reviewer 1.

Comment 28. *Figure 6(b) and (c). Are these percentages a quantile in multidimensional space or something else? This is not clear from either the text or the figure.*

Response: Thank you for the comment. We revised the Results section to clarify:

To evaluate the coverage achieved by our method, we assess its ability to capture corner cases, as illustrated in Figure R.5. **For each test case, we compute its multidimensional distance from the centroid of the candidate test case pool. Larger distances indicate a greater likelihood of being a corner case.**

We included the following paragraph in the supplementary material to provide a detailed explanation of the calculation process:

In Figure R.5, the percentages are calculated based on each case’s relative distance to the centroid of the candidate pool in multidimensional space. First, we compute the centroid as $\bar{\mathbf{x}} = \sum_{i=1}^N \mathbf{x}_i$. Second, for each case \mathbf{x}_i , we calculate its squared Euclidean distance to the centroid as $D_i = \|\mathbf{x}_i - \bar{\mathbf{x}}\|_2^2$, yielding distances D_1, \dots, D_N . We then determine the p -th quantile threshold q_p of the set D_i . Third, for each selected case z_i , we compute $D_i^s = \|z_i - \bar{\mathbf{x}}\|_2^2$. Finally, we count how many of the M selected cases satisfy $D_i^s > q_p$.

Comment 29. *Page 9: “If the relative distance between any two adjacent points are far away each other, then they can have higher chance to cover the entire data space”. What does it mean a higher chance? The data space limits are known because they are defined by the larger pool. The chance to cover the entire data space is deterministic.*

Response: Thank you for the comment. We revised the sentence as follows:

If the relative distances between adjacent points are large, **the corresponding point configuration exhibits better space-filling properties, thereby providing more comprehensive coverage of the data space (Wang et al., 2022; Joseph, 2016).**

Comment 30. *Equation 5: the double sum is not equal to the variance. There is a $2x$ factor.*

Response: Thank you for the comment. We updated the mentioned equation as follows:

This relative distance among the M points z_1, \dots, z_M can be expressed as:

$$\frac{1}{2M(M-1)} \sum_{i=1}^M \sum_{j \neq i}^M (z_i - z_j)^2 = \frac{1}{M-1} \sum_{i=1}^M (z_i - \bar{z})^2 = \text{Variance} \quad (\text{R.3})$$

Comment 31. *Page 10: “Smaller values of MMD_2 can indicate the selected cases z_1, \dots, z_M closely approximate the distribution”. What does it mean that they can? That they also cannot?*

Response: Thank you for the comment. According to Gretton et al. (2012), in expectation, a lower MMD_2 value corresponds to a smaller discrepancy between the empirical distribution of the selected cases z_1, \dots, z_M and the reference distribution. We revised the sentence in the manuscript as follows:

Smaller values of $MMD_u^2(\mathbb{P}_x, \mathbb{P}_z)$ indicate that the selected cases z_1, \dots, z_M closely approximate the target distribution \mathbb{P}_x .

Comment 32. *Page 10: “With the importance measures w_1, \dots, w_N , the optimization implemented in step 1 from Equation 15 should be equivalent as $\min IP_{x, wgt}$ ”. “Should be equivalent” gives the impression that the authors are not sure.*

Response: Thank you for the comment. We revised the sentence as follows:

the optimization implemented in Step 1 is equivalent to solving $\min IP_{x, wgt}$

Comment 33. *The code only seems to reproduce figures and tables, but I could not find the algorithms described in the paper, in particular the KTCS algorithm.*

Response: Thank you for the comment. We included example data and the corresponding expected output of the KTCS algorithm in the resubmission.

Response to the reviewer 2’s comments

Ensuring the safety of Automated Driving Systems (ADS) requires a thorough and standardized approach to selecting test cases that encompass both common (representative) and rare (high-risk) driving scenarios. The authors identify several key challenges. First, the complexity of real-world driving makes it difficult to design comprehensive test scenarios, since real driving involves numerous factors such as vehicle dynamics, road conditions, and interactions with other vehicles. Second, the rarity of safety-critical events—accidents and near-crashes—means that vast amounts of driving data would be necessary to capture them if one relied solely on random observation in normal conditions.

To address these challenges, the paper leverages data from the Second Strategic Highway Research Program (SHRP2)—the largest-scale naturalistic driving study (NDS) in the United States. From this database, the authors aim to systematically select a small but strategically chosen set of test cases that reflects the distribution of real-world driving while also including more extreme corner cases.

They introduce Kernel Test Case Sampling (KTCS), a two-step algorithm to select a manageable subset of test cases from the large SHRP2 pool. The algorithm seeks to balance coverage (ensuring that sparse or corner-case scenarios are included) with representativeness (ensuring that the selected subset mirrors the overall driving distribution). Since the selection process draws on naturalistic driving data, the resulting test cases reflect actual behaviors, speeds, distances, and accident rates.

Additionally, the paper proposes a novel safety measure called Scaling Risk, which compares an ADS’s crash rate directly to that of human drivers in SHRP2. This comparison helps close a gap in existing testing methods: government and industry generally benchmark ADS safety against typical human crash rates. The authors also provide theoretical analysis showing that, given a sufficiently large data pool, the weighted subset converges to an accurate representation of the underlying distribution. This “asymptotic” justification suggests that a small, well-chosen sample can remain statistically valid.

Comment 1. *Despite these contributions, some limitations remain. The authors point out that using generative models to create additional corner cases not present in the NDS data could help cover extremely rare situations, such as multi-vehicle collisions. Moreover,*

although the paper focuses on simulation and test-track environments, a potential future step would be to extend these 118 test cases to more complex virtual or real-world settings. In doing so, a robust library of field-validated scenarios could serve as a shared reference for both industry and regulators.

Response: Thank you for the comment. We revised the Discussion section, with all changes marked in red.

Several potential extensions can be explored in future studies. Building on these insights, recent advances in deep generative modeling can be leveraged to create rare or unseen driving scenarios **and to sample variations around the identified failure cases (Russell et al., 2025; Guan et al., 2024).** By amplifying these challenging situations, generative models can produce a dense cloud of near-miss trajectories, and testing the ADS against this expanded set yields stronger statistical evidence when comparing its safety performance to that of human drivers. **Encoding these test cases in a standard format, such as OpenSCENARIO (ASAM, 2024), will enable their integration into a growing, field-validated library that can be shared across industry and regulatory agencies.** Insights gained from these evaluations can guide more rigorous simulations and field trials on industry-standard platforms (Yan et al., 2023), supporting scalable validation and targeted fine-tuning of ADS performance in controlled yet dynamic virtual environments or on dedicated test tracks. Researchers can also leverage the selected testable cases to inform the development of rigorous ADS safety test protocols, using industry-standard software or constructing field operation tests on dedicated test tracks.

Comment 2. *A final challenge is the scalability of kernel-based methods: these can become computationally expensive for very large N . Although the paper presents approximations and theoretical insights, it generally presupposes that certain efficiency or approximation strategies can be used for big data.*

Response: Thank you for the comment. We revised the Discussion section to discuss the scalability of kernel methods:

The proposed KTCS algorithm is dataset-agnostic and can be applied to a wide range of naturalistic driving studies. We build test cases from strategically sampled subsets of the SHRP2 NDS. Future research could scale this methodology to the full SHRP2 database or apply it to other naturalistic driving studies and industry crowdsourcing initiatives. Doing so would capture evolving traffic patterns and vehicle technologies, and refining human-performance baselines for modern ADS evaluations. These extensions could expand the pool of candidate test cases to hundreds of millions.

At such scale, standard kernel operations become computationally expensive. Recent advances in statistical computing offer viable solutions, such as low-rank approximations to compress kernel functions (Si et al., 2017), or mini-batch kernel algorithms that partition data into blocks for parallel processing (Teymur et al., 2021). These strategies can make large-scale kernel analyses computationally feasible.

Comment 3. *Explain the use of Accident-Rate-SUT as a performance metric.*

Response: Thank you for the comment. We revised the Results Section in the manuscript:

Given a driving system under test (SUT), it is evaluated on a set of 118 selected test cases. For each test case j , we define a binary indicator $I_j \in \{0, 1\}$, where $I_j = 1$ indicates that the SUT passed the case, and $I_j = 0$ indicates a failure (e.g., a crash). To evaluate the safety performance of the SUT, we introduce **Accident-Rate-SUT**, a metric that converts the binary test outcomes (pass/fail) of the representative test cases into a real-world, safety-oriented measure. This metric is motivated by the *golden rule* in automated driving system (ADS) safety research: accident rate, defined as the number of crashes divided by the total driving distance (Feng et al., 2021, 2023). The exposure distance d_j for each case is calculated as the product of the average speed and the fixed test duration (15 seconds). Our method employs an attention mechanism to assign importance weights $\lambda_1, \dots, \lambda_M$ to each test case. Details of the selected test cases, their scenario features, and the corresponding weights are provided in the

Supplementary Material.

The Accident-Rate-SUT is designed to quantify the SUT’s safety performance by aggregating failures across test scenarios in a risk-sensitive manner. Specifically, we sum the number of failed test cases (i.e., crashes observed in the selected test scenarios), each weighted by its corresponding importance score as computed in Step 2 of the KTCS algorithm. This weighted sum is then normalized by the cumulative exposure distance of all test cases. The resulting Accident-Rate-SUT reflects the expected number of crashes per meter traveled. As shown in Figure 7, if the SUT fails on test case IDs [15, 12, 104, 111, 28], the corresponding accident rate is approximately 2.352×10^{-8} crashes per meter.

Comment 4. *The Discussion section (page 8) is somewhat weak; stronger elaboration on the results would improve clarity.*

Response: Thank you for the comment. We kindly refer you to our responses to Comments 1 and 2 from Reviewer 2 for further details.

Comment 5. *Algorithm 1 (page 11) needs more explanation.*

Response: Thank you for the comment. The explanations for Algorithm 1 are directly quoted as the follows.

Based on the optimized function $w_{\theta_{\text{opt}}}(\cdot)$ for $\mathbf{x}_1, \dots, \mathbf{x}_N$, we generate importance measures $w_{\theta_{\text{opt}}}(\mathbf{x}_1), \dots, w_{\theta_{\text{opt}}}(\mathbf{x}_N)$. Pareto order sampling is then applied to generate exactly M sub-samples $\mathbf{z}_1, \dots, \mathbf{z}_M$ based on these importance measures. A more detailed discussion of Pareto order sampling can be found in Tillé (2020). We first generate random variables U_1, \dots, U_N from a Uniform(0,1) distribution. Then, for each case i , we compute the Pareto ranking variables as $Q_i = \frac{U_i}{1-U_i} \times \frac{1-w_{\theta_{\text{opt}}}(\mathbf{x}_i)}{w_{\theta_{\text{opt}}}(\mathbf{x}_i)}$ for $i = 1, \dots, N$. Among all Q_1, \dots, Q_N , we select the M smallest values. The corresponding M samples are then selected as the final sub-sample, weighted according to their importance measures $w_{\theta_{\text{opt}}}(\mathbf{x}_1), \dots, w_{\theta_{\text{opt}}}(\mathbf{x}_N)$.

We further added the explanation of the KTCS algorithm to the revised manuscript. These revisions are marked in red.

KTCS algorithm has two steps. Step 1 focuses on a discrete design, where the key idea is to select M cases from the original pool of N candidate test case. The optimization variables are w_1, \dots, w_N for $\mathbf{x}_1, \dots, \mathbf{x}_N$, and the objective is to minimize the IP. Minimizing IP promotes well-dispersed selections in the feature space, serving as a surrogate for diversity and coverage. In Step 2, we re-weight the M selected cases so that their empirical distribution aligns as closely as possible with that of the full candidate pool. The optimization variables are $\lambda_1, \dots, \lambda_M$ for $\mathbf{z}_1, \dots, \mathbf{z}_M$, and the objective is to minimize MMD, which quantifies representativeness. Together, this dual-step process creates a computationally efficient framework that selects and weights data points to maintain both diversity and representativeness, enabling accurate and scalable inference.

Comment 6. *In the theoretical discussion (page 12), consider providing a brief proof to enhance transparency.*

Response: Thank you for the comment. The proofs of Theorem 1 to Theorem 3 are provided on pages 10 to 20 of the supplementary materials.

Response to the reviewer 3's comments

In this paper, the authors present an algorithm for sampling test cases from a large test pool for the testing of autonomous driving systems. Experimental results demonstrate that the test cases sampled by the algorithm have high representativeness as well as coverage.

Comment 1. *One major issue of the paper is the logic of presentation. The methodology is presented as the last section of the paper (page 9 to 13), whereas results are given right after Introduction (page 3 to 8). Discussion is provided in between (page 8 to 9). No conclusion is provided. Such a structure of the manuscript makes it very difficult for readers to understand the work.*

Response: Thank you for the comment. Per your suggestion, we reorganized the structure in the revised manuscript. Below is an outline of the main sections and subsections with their associated page numbers:

- 1. Introduction (Page 1)
- 2. Methods (Page 4)
 - 2.1 Evaluation Metrics (Page 4)
 - 2.2 Stochastic Sampling Algorithm (Page 5)
 - 2.3 Attention-Based Mechanism (Page 6)
 - 2.4 KTCS Algorithm (Page 7)
 - 2.5 Theoretical Discussion (Page 7)
- 3. Results (Page 8)
 - 3.1 Data (Page 8)
 - 3.2 Experiment Setup (Page 8)
 - 3.3 Performance Comparison (Page 9)
 - 3.4 Statistical Realism (Page 11)
 - 3.5 Demonstration (Page 13)
- 4. Conclusion and Discussion (Page 14)

Comment 2. *In terms of technical details, it is surprising to see “single-vehicle conflicts are excluded” as the authors conjecture that “they primarily result from human errors.” As a matter of fact, the current automated driving models are not immune from single-vehicle conflicts; for example, there are reports on single-vehicle conflicts caused by Tesla’s Autopilot. The authors must provide strong rationale behind this setting (excluding single-vehicle conflicts).*

Response: Thank you for the comment. The SHRP2 NDS utilizes millimeter-wave radar to detect surrounding vehicles (Hankey et al., 2016). All risk surrogate features related to the surrounding environment that we extract, such as *xpos-min* (the minimum longitudinal gap to the vehicle in front), require at least one other moving object to calculate relative motion. In cases of single-vehicle incidents (such as road departures or impacts with fixed

objects), no other moving vehicles can be detected by the radar systems used in the SHRP2 NDS. As a result, we cannot extract radar-based features for these incidents, and including them would increase missing values in the dataset.

Our aim is to model interaction-induced crash risk, which emerges from conflicts between two or more road users. Interaction-induced crashes are the primary goal of ADS safety evaluations (Feng et al., 2021; Yan et al., 2023). Single-vehicle crashes arise from different mechanisms, such as distraction, impairment, loss of control (Kassing and Gibbons, 2024). To avoid conflating these fundamentally different processes, we limit the analysis to multi-party conflicts. Accordingly, we added the following discussion in the revised manuscript:

Single-vehicle conflicts were excluded because, when no other moving vehicles are present, the radar cannot provide the surrounding-environment data needed to compute surrogate risk metrics (Hankey et al., 2016).

Comment 3. *It may be an overclaim to say “our approach effectively achieves both representativeness and coverage objectives” based on the results given in Table 1 (page 6). The approach gives a good trade-off between representativeness and coverage, instead of achieving both of them. From Table 1, we can observe that some baseline technique(s) outperformed the proposed approach in some metrics, while other baselines outperformed in other metrics. The proposed approach just gave a good balance among these metrics.*

Response: Thank you for the comment. To clarify the statement, we revised the Results section as follows:

In summary, Table 1 shows **our method achieves competitive scores on both IP and MMD, demonstrating balanced performance across these metrics. Our method outperforms MMD-Critic by achieving lower values on both measures. MED fails to meet the representativeness objective, as its MMD is substantially higher than that of all other methods. Although SP-Uniform and SPARTAN achieve slightly lower MMD values, they produce substantially higher IP scores (all above 0.360).**

Comment 4. *For the results of statistical realism, can quantitative measurement be*

given? Currently, the realism is visualized (through Figure 5 and 6) in a qualitatively way. It would also be worthwhile to compare the presented approach’s realism with those of baseline techniques, like what have been done for the results in “Performance Comparison” (Figure 3 and 4, and Table 1).

Response: Thank you for this insightful comment. Figure 5 originally provided a qualitative impression of how well our selected test cases reproduce the distribution of all 48 investigated features in the full candidate test case pool. In response to the reviewer’s recommendation, we introduced a quantitative measure of *statistical realism* and compared our method against all baseline techniques, similar to the evaluation performed in the Performance Comparison section (Figure 3, Figure 4, and Table 1).

To this end, we added the following analysis to the Results section and updated Figure 5 accordingly:

Similarly, we quantitatively compare the per-feature discrepancy across all methods using a heatmap in Figure R.4, where the per-feature discrepancy is measured by $ADD_j = \frac{1}{N} \sum_{i=1}^N |\hat{p}_j(\mathbf{x}_i) - p_j(\mathbf{x}_i)|$. Here, ADD_j represents the contribution of feature j to the overall ADD, The global ADD reported in the main paper is the sum of these 48 per-feature discrepancy values. Figure R.4 consists of eight sub-panels: the first row displays all kinematics-based features, and the second row shows all radar-based features. Darker cells indicate larger discrepancies, corresponding to lower statistical realism.

We revised Figure 6 and added the following analysis to the Results section to provide quantitative results:

Figure R.5(c) presents a quantitative comparison of long-tail scenario coverage across all baseline methods. The results show that Uniform, SP, and SPARTAN offer limited coverage of long-tail scenarios, as only a few of their selected cases exceed the 99% percentile. The MED method captures the tail almost exclusively, with 117 of 118 selected cases exceeding the 95% percentile. Such strong emphasis on extreme values limits its ability to represent the full distribution, thereby

Figure R.4: **Heat-map of per-feature discrepancies across different methods.** The first row presents results for kinematics-based features, while the second row presents results for radar-based features.

reducing its overall statistical representativeness.

Figure R.5: **Schematic diagram of coverage evaluation.** (a) Illustration of distance calculations for the central density region. (b) Identification of corner cases based on varying quantile thresholds. (c) Summary of corner-case coverage across varying quantile thresholds.

Comment 5. *The code is just to generate some figures/table. It is not so relevant to the technical details of the proposed algorithm.*

Response: Thank you for the comment. We included example input data and the corresponding expected output of the KTCS algorithm in the resubmission.

Response to the reviewer 4's comments

Other existing or under development safety argumentation frameworks deep dive into the approach itself, where scenario sampling is just one step. For instance, the Safety Assurance Framework (SAF) of the European research project SUNRISE, proposes a variety of steps that go far beyond scenario sampling. These include the selection of abstract and logical scenarios, the concretization of scenarios, the allocation of scenarios into test instances, the preparation of test cases (including the scenario + the function under test), the execution of tests (in virtual and real conditions), the evaluation of the results, and the potential consideration of in-service monitoring. In comparison, this paper’s argumentation, whose main focus is the optimization of test case sampling, lacks a broader lifecycle management of the safety of ADAS/ADS functions. As a consequence, in my opinion, it would be difficult to consider the methodology proposed in this paper as a complete and comprehensive solution for managing the safety lifecycle of ADAS/ADS functions, especially in complex contexts such as the European automotive ecosystem, also considering the geographical and temporal limitations of the underlying SHRP2 NDS dataset. That said, the method proposed fits very well with certain steps of the SUNRISE SAF, providing a sound and efficient mechanism to select (sample) scenarios to be executed in the test instances, which could be a valuable tool within the scenario selection or test case preparation phase of the SAF.

Response: We appreciate the reviewer’s comment regarding the comparison between our approach and the Safety Assurance Framework (SAF) of the European research project SUNRISE. We would like to take this opportunity to clarify the scope of our work.

The SHRP2 NDS is a U.S. nationwide naturalistic driving study, designed as a national survey research. Such studies are systematically structured to collect data representative of the U.S. driving population. SHRP2 NDS employs stratified, multi-stage probability sampling across six geographically diverse sites (Dingus et al., 2016). The dataset captures a wide range of driver demographics, environmental conditions, and vehicle types (Dingus et al., 2015).

A key distinction from SAF SUNRISE is that SHRP2 NDS enables researchers to associate scenario occurrences with human driving behavior under natural exposure conditions, allowing for evaluation of safety risks under nationwide representation, rather than treating each scenario as a predefined test case. The selected test case pool can thus be used to assess the

safety performance of automated systems in direct comparison to human drivers facing the same circumstances.

Rather than focusing on outcomes from individual test cases, our method leverages distributional information from large-scale NDS data to reveal system-level behavioral patterns that may not emerge through isolated testings. Without such a human-grounded baseline, commonly used risk metrics, such as crash rate, may lack the necessary context to assess whether ADS performance is socially and operationally acceptable. Incorporating this comparative benchmark enhances both the interpretability and real-world relevance of safety validation efforts, including those structured under SAF.

In our view, the proposed method and SAF are complementary. The proposed algorithm provides a foundation for benchmarking and risk-informed scenario selection grounded in real-world human behavior.

We thank the reviewer again for this insightful comment.

Comment 1.1 *Regarding the validity of the SHRP2-NDS. How might the dates when the SHRP2 NDS data was collected (2010-2013) impact its current relevance (2025) for ADS validation, given advancements in vehicle technology and availability of ADAS in modern cars?*

Response: Thank you for the comment. We believe that the SHRP2 NDS remains a valuable resource for evaluating today’s ADS technologies. The primary goal of utilizing this dataset is to assess ADS risk by comparing its behavior with that of human drivers under naturalistic conditions, rather than to replicate the specific vehicle technologies from 2010 to 2013. SHRP2 serves as a proof-of-concept foundation due to its extensive scale and design rigor. Importantly, the proposed KTCS algorithm is dataset-agnostic and can be readily integrated with other scenario-based datasets or testing environments where accelerated evaluation of automated systems is required.

In response to the reviewer’s suggestion, we revised the Discussion Section to clarify its relevance and potential for future research:

The proposed KTCS algorithm is dataset-agnostic and can be applied to a wide range of naturalistic driving studies. We build test cases from strategically sam-

pled subsets of the SHRP2 NDS. Future research could scale this methodology to the full SHRP2 database or apply it to other naturalistic driving studies and industry crowd-sourcing initiatives. Doing so would capture evolving traffic patterns and vehicle technologies, and refining human-performance baselines for modern ADS evaluations.

Comment 1.2 *To what extent do the driving behaviors and traffic scenarios within SHRP2 NDS still accurately reflect present-day real-world driving conditions?*

Response: Thank you for the comment. While the evolution of national driving behaviors and traffic scenarios falls within the broader scope of the SHRP2 NDS, it lies somewhat outside the primary focus of our work. We find no strong evidence that aggregate-level driving behavior or the distribution of traffic scenarios has changed significantly. Similarly, there is limited evidence to suggest that aggregate risk estimates (Bureau of Transportation Statistics, 2024) have undergone substantial changes over time. These estimates include crash rates per vehicle mile traveled or risk levels associated with different scenario types (Stewart, 2022, 2023). We would be happy to further explore this point if the reviewer could clarify any specific aspects or provide relevant references of concern.

Comment 1.3 *Given that SHRP2 NDS represents US driving characteristics, are there limitations to applying scenarios from this dataset to evaluate ADS in European or other non-US driving environments?*

Response: Thank you for the comment. The naturalistic driving data used in our study includes traffic scenarios such as car-following (Hammit et al., 2018), lane changing (Xu et al., 2024), and intersection negotiation scenarios (Zafian et al., 2021), which are fundamental and commonly encountered across diverse regions. These scenario types serve as essential building blocks in defining and designing operational design domains (ODDs) for ADS and are generally applicable beyond any specific geographic context.

Comment 1.4 *Are these SHRP2 NDS aspects anyhow affecting the proposed KTCS approach? (or it is completely unrelated?)*

Response: Thank you for the question. The specific design aspects of the SHRP2 NDS do not affect the proposed KTCS approach. KTCS is a dataset-agnostic algorithm that op-

erates on abstracted scenario features and their associated risk estimates. The algorithm is applicable to any structured driving dataset that offers comparable contextual information and is thus not constrained by the specific characteristics or design choices of the SHRP2 NDS.

2. Overall comments

The paper is of high quality in terms of formatting and language, showcasing careful proof-reading and editing. All the references and figures within the document are consistent and logically integrated throughout the text, which greatly help the reader to understand the message.

Figures are detailed and built as rich infographics that help the reader to understand concepts easily.

The code is provided as an online script with all the data necessary to reproduce the results and figures. It includes Python routines to load the data, extract the statistics, and plot the results. It also contains a version history of previous results. Overall, the code presented supports well the argumentations on the paper.

Response: We sincerely appreciate your encouraging remarks and positive feedback.

References

Albrecht, H., Barickman, F. S., Schnelle, S. C., et al. (2021). Advanced test tools for adas and ads. Technical report, United States. Department of Transportation. National Highway Traffic Safety Administration, DOT HS 813 083.

ASAM (2024). Open scenario. <https://www.asam.net/standards/detail/openscenario/v200/>.

Bureau of Transportation Statistics (2024). Transportation statistics annual report 2024. Technical report, U.S. Department of Transportation.

- Chai, J. and Wang, X. (2022). Fairness with adaptive weights. In *International Conference on Machine Learning*, pages 2853–2866. PMLR.
- Chen, Y., Song, S., Li, S., and Wu, C. (2019). A graph embedding framework for maximum mean discrepancy-based domain adaptation algorithms. *IEEE Transactions on Image Processing*, 29:199–213.
- Dingus, T. A., Guo, F., Lee, S., Antin, J. F., Perez, M., Buchanan-King, M., and Hankey, J. (2016). Driver crash risk factors and prevalence evaluation using naturalistic driving data. *Proceedings of the National Academy of Sciences*, 113(10):2636–2641.
- Dingus, T. A., Hankey, J. M., Antin, J. F., Lee, S. E., Eichelberger, L., Stulce, K. E., McGraw, D., Perez, M., and Stowe, L. (2015). *Naturalistic driving study: Technical coordination and quality control*. Number SHRP 2 Report S2-S06-RW-1.
- Feng, S., Sun, H., Yan, X., Zhu, H., Zou, Z., Shen, S., and Liu, H. X. (2023). Dense reinforcement learning for safety validation of autonomous vehicles. *Nature*, 615(7953):620–627.
- Feng, S., Yan, X., Sun, H., Feng, Y., and Liu, H. X. (2021). Intelligent driving intelligence test for autonomous vehicles with naturalistic and adversarial environment. *Nature Communications*, 12(1):748.
- Gretton, A., Borgwardt, K. M., Rasch, M. J., Schölkopf, B., and Smola, A. (2012). A kernel two-sample test. *The Journal of Machine Learning Research*, 13(1):723–773.
- Guan, Y., Liao, H., Li, Z., Hu, J., Yuan, R., Zhang, G., and Xu, C. (2024). World models for autonomous driving: An initial survey. *IEEE Transactions on Intelligent Vehicles*.
- Guo, F. (2019). Statistical methods for naturalistic driving studies. *Annual Review of Statistics and its Application*, 6:309–328.
- Hammit, B. E., Ghasemzadeh, A., James, R. M., Ahmed, M. M., and Young, R. K. (2018). Evaluation of weather-related freeway car-following behavior using the shrp2 naturalistic driving study database. *Transportation Research Part F: Traffic Psychology and Behaviour*, 59:244–259.

- Han, R., Shi, P., and Zhang, A. R. (2024). Guaranteed functional tensor singular value decomposition. *Journal of the American Statistical Association*, 119(546):995–1007.
- Hankey, J. M., Perez, M. A., and McClafferty, J. A. (2016). Description of the shrp 2 naturalistic database and the crash, near-crash, and baseline data sets. Technical report, Virginia Tech Transportation Institute.
- Joseph, V. R. (2016). Space-filling designs for computer experiments: A review. *Quality Engineering*, 28(1):28–35.
- Kassing, A. and Gibbons, R. B. (2024). Roadway departure events using shrp 2 nds data. Technical report, National Surface Transportation Safety Center for Excellence.
- Li, C.-L., Chang, W.-C., Cheng, Y., Yang, Y., and Póczos, B. (2017). Mmd gan: Towards deeper understanding of moment matching network. *Advances in Neural Information Processing Systems*, 30.
- Muandet, K., Fukumizu, K., Sriperumbudur, B., Schölkopf, B., et al. (2017). Kernel mean embedding of distributions: A review and beyond. *Foundations and Trends[®] in Machine Learning*, 10(1-2):1–141.
- Pourahmadi, M. (2013). *High-dimensional covariance estimation: with high-dimensional data*. John Wiley & Sons.
- Principe, J. C. (2010). *Information theoretic learning: Renyi’s entropy and kernel perspectives*. Springer Science & Business Media.
- Russell, L., Hu, A., Bertoni, L., Fedoseev, G., Shotton, J., Arani, E., and Corrado, G. (2025). Gaia-2: A controllable multi-view generative world model for autonomous driving. *arXiv preprint arXiv:2503.20523*.
- Scanlon, J. M., Kusano, K. D., Daniel, T., Alderson, C., Ogle, A., and Victor, T. (2021). Waymo simulated driving behavior in reconstructed fatal crashes within an autonomous vehicle operating domain. *Accident Analysis & Prevention*, 163:106454.

- Si, S., Hsieh, C.-J., and Dhillon, I. S. (2017). Memory efficient kernel approximation. *Journal of Machine Learning Research*, 18(20):1–32.
- Simon-Gabriel, C.-J. and Schölkopf, B. (2018). Kernel distribution embeddings: Universal kernels, characteristic kernels and kernel metrics on distributions. *Journal of Machine Learning Research*, 19(44):1–29.
- Stewart, T. (2022). Overview of motor vehicle crashes in 2020. Technical report, United States. Department of Transportation. National Highway Traffic Safety Administration, DOT HS 813 266.
- Stewart, T. (2023). Overview of motor vehicle crashes in 2021. Technical report, United States. Department of Transportation. National Highway Traffic Safety Administration, DOT HS 813 435.
- Teymur, O., Gorham, J., Riabiz, M., and Oates, C. (2021). Optimal quantisation of probability measures using maximum mean discrepancy. In *International Conference on Artificial Intelligence and Statistics*, pages 1027–1035. PMLR.
- Tillé, Y. (2020). *Sampling and estimation from finite populations*. John Wiley & Sons.
- Vaswani, A., Shazeer, N., Parmar, N., Uszkoreit, J., Jones, L., Gomez, A. N., Kaiser, L., and Polosukhin, I. (2017). Attention is all you need. *Advances in Neural Information Processing Systems*, 30.
- Wang, Y., Sun, F., and Xu, H. (2022). On design orthogonality, maximin distance, and projection uniformity for computer experiments. *Journal of the American Statistical Association*, 117(537):375–385.
- Wolfer, G. and Alquier, P. (2025). Variance-aware estimation of kernel mean embedding. *Journal of Machine Learning Research*, 26(57):1–48.
- Xu, J., Qian, C., Han, S., and Guo, F. (2024). Detecting critical mismatched driver visual attention during lane change: An embedding kernel algorithm. *IEEE Transactions on Intelligent Transportation Systems*, 25(7):7070–7080.

- Yan, X., Zou, Z., Feng, S., Zhu, H., Sun, H., and Liu, H. X. (2023). Learning naturalistic driving environment with statistical realism. *Nature Communications*, 14(1):2037.
- Zafian, T., Ryan, A., Agrawal, R., Samuel, S., and Knodler, M. (2021). Using shrp2 nds data to examine infrastructure and other factors contributing to older driver crashes during left turns at signalized intersections. *Accident Analysis & Prevention*, 156:106141.
- Zhang, J., Meng, C., Yu, J., Zhang, M., Zhong, W., and Ma, P. (2023). An optimal transport approach for selecting a representative subsample with application in efficient kernel density estimation. *Journal of Computational and Graphical Statistics*, 32(1):329–339.

Response to comments

“Optimizing Test Case Sampling for Safety Validation of Automated Driving System with Naturalistic Driving Study” (NCOMMS-25-16414A)

We express our sincere gratitude for reviewers’ valuable and constructive comments. We have carefully revised the paper based on the comments and presented a point-by-point response addressing each comment. The comments are presented in italics, followed by our response. Please note that all page numbers referenced in this response correspond to the revised version of the manuscript.

Response to the reviewer 1’s comments

Comment 1. *Thank you for addressing my previous points. My only remaining major point concerns the optimised function $w_{\theta_{opt}}$, where theta is a p -dimensional vector. Both the function and theta are discussed in the Methods section, but are not mentioned again in either the Results section or the Supplementary Material. What is the value of p in practice? (I suppose $p \ll N$). Which theta vector has been estimated from the data? Does the estimated theta vector provide any information, or do we simply have to trust it? It is also unclear whether p can be any number or if it must be selected within a given range.*

Response: Thanks for the positive feedback. Regarding the weight function w_{θ} , we added the following content in the supplementary material to clarify, as quoted below.

Following the framework of Ren et al. (2018); Zhao et al. (2019), we define the weight function as $w_{\theta}(\mathbf{x}) = c[E(\mathbf{x})^{\top}\theta]^2$ where c is a normalization constant ensuring that $\sum_{i=1}^N w_{\theta}(\mathbf{x}_i) = 1$. In this context, $E(\mathbf{x})$ functions as a dimension-reduction operator that projects the original input features from dimension d to p , with p denoting the effective feature dimension (Dai et al., 2022) and d representing the dimensionality of the original input. We parameterize $E(\mathbf{x})$ using a single-layer neural network with a ReLU activation function, and θ is

specified as a $p \times 1$ vector that maps the embedded representation to a scalar weight output.

For the choice of dimensionality p , Theorem 2 indicates that to maintain consistency, p must be substantially smaller relative to the sample size ($p \ll N$). In practice, it is reasonable to require the effective dimension p to be smaller than the dimension of the input features ($p < d$). Following Zhao et al. (2019), an ideal p lies in $[10, 30]$ and we set $p = 16$ in the numerical experiments. The parameters θ and the embedding function $E(\cdot)$ are learned by minimizing Equation 8 using the Adam optimizer (Kingma and Ba, 2014). We set the number of iterations to 200 and the learning rate to 5×10^{-4} to ensure sufficient convergence. Importantly, the primary output of interest is not the θ and embedding function, but the weights it produces.

Comment 2. *Page 3: SUT is used without being defined. I also believe SUT and ADS are used to refer to the same thing. At least in page 3.*

Response: Thanks for the comment. SUT refers to the System Under Test. You are correct that in this context the SUT is indeed the ADS to be tested:

To demonstrate the interpretation of the test results using selected cases, a measure called *scaling risk* is proposed to quantify the safety level of an ADS system’s performance based on the success or failure of each case, where a case is labeled a success if the **system under test (SUT)** completes it without a traffic crash, and a failure if the ADS does not complete it safely and a crash occurs.

Comment 3. *d must be defined soon after \mathbb{R}^d and not in the following paragraph.*

Response: Thanks for the comment. We reallocated the definition of d in the text as follows:

With appropriate data pre-processing, let $\mathbf{x}_1, \dots, \mathbf{x}_N$ represent the test case pool and each data point $\mathbf{x}_i \in \mathbb{R}^d$, **where d denotes the dimension of the features for each case.**

Comment 4. *The sentence “Ideally, this subset z_1, \dots, z_M contains both common cases and corner cases, as entire sets, the probability distribution function, constructed based on the selected subsets z_1, \dots, z_M , approximates the distribution of x_1, \dots, x_N ” is not clear and has some problems with the English. Please consider to use two sentences. I would also remove “Ideally” because that must happen in practice and not (only) ideally.*

Response: We appreciate the comment. The sentence has been revised accordingly, shown as follows:

The selected subsets z_1, \dots, z_M should collectively include both common cases and corner cases. In addition, to ensure representativeness, the probability distribution function constructed from these subsets should closely approximate the empirical distribution of the original sample x_1, \dots, x_M .

Comment 5. *The sentence “Consider in space X ” is not clear or wrong. Maybe the “in” should be removed?*

Response: Thanks for the comment. We have removed the word “in” to improve clarity, together with other revision. The text is provided in the response to Comment 6 below.

Comment 6. *What kind of space is X ? Please clearly define things the first time they are introduced.*

Response: We added the clarification for \mathbb{X} . Specifically, \mathbb{X} is the feature space where the test cases reside. Each data point $x_i \in \mathbb{X} \subseteq \mathbb{R}^d$ represents a case. We revise the corresponding content in the manuscript as the follows:

We use kernel methods to measure the two objectives in this study: representativeness and coverage. Consider a feature space $\mathbb{X} \subseteq \mathbb{R}^d$, where each test case $x_i \in \mathbb{X}$. Define \mathcal{K} as a kernel function mapping $\mathbb{X} \times \mathbb{X} \rightarrow \mathbb{R}$.

Comment 7. *A “)” is missing in the exponential formula (also elsewhere)*

Response: Thanks for the comment. We revise this as the follows:

For example, the Radial basis function (RBF) kernel $\mathcal{K}(z_t, z_s) = \exp\left(\frac{-\|z_t - z_s\|_d^2}{2\sigma^2}\right)$

is a popular one, where the parameter σ denotes the bandwidth of the RBF kernel.

We revise another formula as the follows:

For example, if we are using the RBF kernel as $\mathcal{K}(\mathbf{z}_t, \mathbf{z}_s) = \exp\left(\frac{-\|\mathbf{z}_t - \mathbf{z}_s\|_d^2}{2\sigma^2}\right)$, then $\mathbb{E}_{\mathbf{z} \sim \mathbb{P}}[\mathcal{K}(\mathbf{z}, \mathbf{z})] = 1$.

Comment 8. *sigma in the exponential formula is not defined.*

Response: Thanks for the comment, we add the explanation for the parameter σ as follows:

For example, the Radial basis function (RBF) kernel $\mathcal{K}(\mathbf{z}_t, \mathbf{z}_s) = \exp\left(\frac{-\|\mathbf{z}_t - \mathbf{z}_s\|_d^2}{2\sigma^2}\right)$ is a popular one, where the parameter σ denotes the bandwidth of the RBF kernel.

Comment 9. *x_i initially belong to R^d but later they belong to X . Does it mean that R^d and X are the same? This is very confusing for the reader.*

Response: As clarified in the response to comment 6 above, $\mathbf{x}_i \in \mathbb{X} \subseteq \mathbb{R}^d$. We also clarified that in the revised text as in quoted text after comment 6.

Comment 10. *The z_i are sometimes bold and some times not.*

Response: Thank you for raising this issue. In our notation, z_i (non-bold) denotes a scalar value, while \mathbf{z}_i (boldface) denotes a vector. Specifically, in Equation (1), z_i refers to a one-dimensional scalar quantity, whereas in other contexts, \mathbf{z}_i represents a multi-dimensional feature vector. We have revised the manuscript to ensure consistent usage and have added an explanatory note to clarify this distinction as quoted below.

Throughout the paper, we use non-bold symbols (e.g., z_i) to denote scalar values and bold symbols (e.g., \mathbf{z}_i) to denote vectors.

Comment 11. *v_z^2 becomes v_z in equation (2) and elsewhere. Is this wanted?*

Response: Thanks for the comment. We revise v_z^2 with v_z the sentence as the follows:

We denote $\nu_{\mathbf{z}} = \mathbb{E}_{\mathbf{z} \sim \mathbb{P}} \|\mathcal{K}(\mathbf{z}, \cdot) - \mu_{\mathbb{P}}\|_{\mathcal{H}}^2$ as the variance for the multivariate \mathbf{z} in the RKHS

Comment 12. *“Here, we use the term P_x ”. Remove term because P_x is not the term of an equation but you are defining it.*

Response: Thanks for the comment. We revise the sentence as follows:

Here, we use $\mathbb{P}_{\mathbf{x}}$ to define the probability distribution constructed based on the points $\mathbf{x}_1, \dots, \mathbf{x}_N$

Comment 13. *“coverage of the data space” (page 4) and “coverage of the data sample” (page 5) are not the same things but they are used like they are. I would use “data space” everywhere.*

Response: Thanks for the comment. We have revised it accordingly on Page 5 as quoted below.

The minimization of the information potential corresponds to maximizing the variance of the feature vectors, which increases the **coverage of data space**.

Comment 14. *MMD is called criterion and metric in two nearby sentences. Please only use metric to avoid confusion with common statistical creteria.*

Response: Thanks for the comment. We revise the corresponding content as follows:

Secondly, we utilize **the Maximum Mean Discrepancy (MMD) as the statistical metric** to measure representativeness.

Comment 15. *The sentence “One way to generate stochastic samplings from the candidate test case pool” gives the idea that the x_i are candidate and can change. But I believe the x_i are fixed.*

Response: Thank you for the insightful comment. To clarify this concern, we have revised the sentence as follows:

Given a fixed candidate test case pool $\mathbf{x}_1, \dots, \mathbf{x}_N$, one way to generate stochastic samplings is to select M points from this pool based on a sequence of assigned importance measures w_1, \dots, w_N .

Comment 16. *In equation (5) S must be s .*

Response: Thanks for the comment. We revise equation (5) as follows, and we highlight the revisions in color red.

In this context, we define the IP with importance measures for the associated points w_1, \dots, w_N as $\text{IP}_{\mathbf{x}, \text{wgt}}$ in the following:

$$\text{IP}_{\mathbf{x}, \text{wgt}} = \sum_{t=1}^N \sum_{s \neq t, s=1}^N w_t w_s \mathcal{K}(\mathbf{x}_t, \mathbf{x}_s) \quad (\text{R.1})$$

Comment 17. *Where are the weights w in step 1 of (11)? Is it correct that they are not used in the double sum?*

Response: Thank you for the comment. The goal of Step 1 in the KTCS algorithm is to select a subset of M cases, $\mathbf{z}_1, \dots, \mathbf{z}_M$, that minimizes the associated Information Potential (IP), as conceptually illustrated in Equation (11). To achieve this minimization in practice, we employ a weighted stochastic sampling procedure, which is detailed in Algorithm 1. The importance measures w_1, \dots, w_N guide the sampling of $\mathbf{z}_1, \dots, \mathbf{z}_M$ from $\mathbf{x}_1, \dots, \mathbf{x}_N$. It is important to note that these w_1, \dots, w_N are part of the sampling method in Algorithm 1 and do not appear directly in the final minimization objective of Equation (11).

We add the following content in the revised manuscript for clarification.

KTCS algorithm has two steps. Step 1 focuses on a discrete design, where the key idea is to select M cases from the original pool of N candidate test case. **The objective function is presented in Step 1 of Equation 11, and the detailed algorithm for achieving this goal is provided in Algorithm 1.**

Comment 18. *“the KTCS method can achieve sufficient representativeness because the constructed density distribution using the selected cases can approximate the original test*

case pool”. The use of “can” gives the idea that those things may or may not happen.

Response: Thanks for the comment. We revise the sentence as follows:

the KTCS method achieves sufficient representativeness because the constructed density distribution using the selected cases approximates well toward the original test case pool.

Comment 19. I also commented on the issues raised by Reviewer #2 as requested by the Editor. Comment 5 from the reviewer is similar to my comment number 3, and I think this is an area where the authors still need to provide more detail. In particular, they should explain which θ was estimated for the specific dataset and how the dimension d was chosen.

Response: Thanks for the comment. The specification of the θ and its associated dimension are discussed in the responses to comment 1. d denotes the dimension of the input features, and they are discussed in Table 4 of the supplementary material.

Response to the reviewer 3’s comments

Comment 1. *Most of my comments in the previous round of review have been well addressed. However, I still have the concern about the exclusion of “single-vehicle conflicts”. It seems that the authors agree with my argument and thus changed the rationale why single-vehicle conflicts were excluded. This is acceptable, but it brings another issue: The authors should explicitly claim the limitation of the scope of the paper, that is, the proposed method is limited to multi-vehicle conflicts only. This limitation should be reflected up front from the paper title.*

Response: Thanks for the comment. Per your suggestion, we have added the following content on page 8 to define the scope of the study. we further justify the focus on multi-vehicle behavioral safety as it constitutes the majority of ADS related crashes. The corresponding content is directly quoted as the follows.

Recent crash data reported by a leading ADS developer indicate that single-vehicle crashes represent around 2% of all ADS-related incidents, with the majority involving multiple vehicles Kusano et al. (2025). Our study focuses on

validating ADS behavioral safety by assessing the response and interactions with surrounding traffic in multi-vehicle involved scenarios (Liu et al., 2025).

We further add the following content to the Discussion section.

Several potential extensions can be explored in future studies. **The current research is intentionally scoped to address behavioral safety in multi-vehicle interaction scenarios. Future work will seek to extend this framework to encompass a broader range of single-vehicle scenarios from a functional safety perspective, thereby advancing toward a comprehensive ADS safety validation framework that integrates both interactive and non-interactive critical events.**

Comment 2. *Although some extra data (example input and expected output) were added, there is still no code showing the implementation of the proposed algorithm.*

Response: Thank you for your valuable feedback. To address your comment, we have released a comprehensive code repository on GitHub, which includes the following:

- A detailed implementation of the KTCS algorithm (see file KTCS-Implementation.ipynb).
- Scripts to reproduce the experimental results presented in the paper.
- A demo illustrating the KTCS’s usage on a sample input (see the file KTCS-Demo.ipynb).

The repository is publicly available at: <https://github.com/tsienchen/KTCS-Project>.

We thank the reviewer again for this insightful comment.

Response to the reviewer 4’s comments

Comment 1. *The authors have addressed all my previous comments and suggestions and added valuable remarks to the manuscript. The authors have debated on my questions on validity of the dataset, applicability in non-US environments, and compatibility with other safety argumentation initiatives. As a result I am convinced the readers will appreciate the*

paper and learn from it.

Response: We sincerely appreciate your encouraging remarks and positive feedback.

Comment 2. *I reviewed the code in the previous review session and I found it complete, clear (with traceability), and supporting the claims. Helpful for further research.*

Response: We sincerely appreciate your encouraging remarks and positive feedback.

References

- Dai, B., Shen, X., and Wang, J. (2022). Embedding learning. *Journal of the American Statistical Association*, 117(537):307–319.
- Kingma, D. P. and Ba, J. (2014). Adam: A method for stochastic optimization. *arXiv preprint arXiv:1412.6980*.
- Kusano, K. D., Scanlon, J. M., Chen, Y.-H., McMurry, T. L., Gode, T., and Victor, T. (2025). Comparison of waymo rider-only crash rates by crash type to human benchmarks at 56.7 million miles. *Traffic Injury Prevention*, pages 1–13.
- Liu, H. X., Yan, X., Sun, H., Wang, T., Qiao, Z., Zhu, H., Shen, S., Feng, S., Stevens, G., and McGuire, G. (2025). Behavioral safety assessment towards large-scale deployment of autonomous vehicles. *arXiv preprint arXiv:2505.16214*.
- Ren, M., Zeng, W., Yang, B., and Urtasun, R. (2018). Learning to reweight examples for robust deep learning. In *International Conference on Machine Learning*, pages 4334–4343. PMLR.
- Zhao, S., Fard, M. M., Narasimhan, H., and Gupta, M. (2019). Metric-optimized example weights. In *International Conference on Machine Learning*, pages 7533–7542. PMLR.